# MathCoder: Seamless Code Integration in LLMs for Enhanced Mathematical Reasoning

**Ke Wang**[1,4][*]   **Houxing Ren**[1][*]   **Aojun Zhou**[1][*]   **Zimu Lu**[1][*]   **Sichun Luo**[3][*]
**Weikang Shi**[1][*]   **Renrui Zhang**[1]   **Linqi Song**[3]   **Mingjie Zhan**[1][†‡]   **Hongsheng Li**[1,2][‡]

[1]Multimedia Laboratory (MMLab), The Chinese University of Hong Kong
[2]Shanghai AI Laboratory   [3]City University of Hong Kong   [4]Nanjing University
{wangk.gm, renhouxing, aojunzhou, zmjdll}@gmail.com
hsli@ee.cuhk.edu.hk

## Abstract

The recently released GPT-4 Code Interpreter has demonstrated remarkable proficiency in solving challenging math problems, primarily attributed to its ability to seamlessly reason with natural language, generate code, execute code, and continue reasoning based on the execution output. In this paper, we present a method to fine-tune open-source language models, enabling them to use code for modeling and deriving math equations and, consequently, enhancing their mathematical reasoning abilities. We propose a method of generating novel and high-quality datasets with math problems and their code-based solutions, referred to as MathCodeInstruct. Each solution interleaves *natural language*, *code*, and *execution results*. We also introduce a customized supervised fine-tuning and inference approach. This approach yields the MathCoder models, a family of models capable of generating code-based solutions for solving challenging math problems. Impressively, the MathCoder models achieve state-of-the-art scores among open-source LLMs on the MATH (45.2%) and GSM8K (83.9%) datasets, substantially outperforming other open-source alternatives. Notably, the MathCoder model not only surpasses ChatGPT-3.5 and PaLM-2 on GSM8K and MATH but also outperforms GPT-4 on the competition-level MATH dataset. The proposed dataset and models will be released upon acceptance.

## 1 Introduction

Recently, closed-source large language models (LLMs) such as GPT-4 (OpenAI, 2023) and PaLM-2 (Anil et al., 2023), paired with methods such as Chain-of-Thought (CoT) (Wei et al., 2022) and Program-Aided Language models (PAL) (Gao et al., 2023), have shown remarkable performance on mathematical reasoning tasks. In contrast, current open-source LLMs (Touvron et al., 2023; Penedo et al., 2023; Zhang et al., 2022) still lag significantly behind in this area. Even Llama-2-70B (Touvron et al., 2023), one of the most potent open-source models, only scores 56.8% and 13.5% respectively on GSM8K (Cobbe et al., 2021) and MATH (Hendrycks et al., 2021) datasets, remarkably lower than GPT-4 Code Interpreter[1], which scores 97% and 69.7% (Zhou et al., 2023a).

To narrow the gap between open-source and closed-source models in math problem solving, recent works, such as the WizardMath (Luo et al., 2023) and RFT (Yuan et al., 2023), have tried to tune open-source models with math problems and CoT solutions, achieving a significant gain in performance compared to their base model, Llama-2. On the other hand, methods such as PAL (Gao et al., 2023), PoT (Chen et al., 2022), and CSV (Zhou et al., 2023a) encourage code usage in solving math problems, showing promising improvements when paired with closed-source models like GPT-3.5, GPT-4 and GPT-4 Code Interpreter. In particular, GPT-4 Code Interpreter surpasses the previous

---

[*]Equal contribution.
[†]Project leader.
[‡]Corresponding author.
[1]https://openai.com/blog/chatgpt-plugins#code-interpreter

**Table 1:** Comparison with different Instruction-following datasets: G and M are the abbreviation for the training subset of GSM8K and MATH dataset. The baseline datsets include recent RFT-u13b (Yuan et al., 2023) and WizardMath (Luo et al., 2023).

| Datasets | Seed | Annotation | Available |
|----------|------|------------|-----------|
| RFT-100k | G | Llama | ✓ |
| WizardMath-96k | G+M | GPT-4 | ✗ |
| Ours-49k | G+M | GPT-4 | ✓ |
| Ours-80k | G+M | GPT-4 + Self-distillation | ✓ |

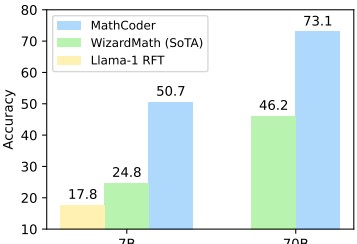

**Figure 1:** Performance comparison between MathCoder, WizardMath, and Llama-1 RFT models with different model sizes.

SOTA by a clear margin. Recent study (Zhou et al., 2023a) shows that this excellent performance can be attributed to its ability to generate and assess the execution results of a chain of code interlaced with natural language reasoning steps. However, existing open-source models fail to benefit from this sophisticated mechanism since they lag behind closed-source models in both code generation and natural language reasoning. *Therefore, we still lack an effective recipe to deliver open-source models to solve math problems in a manner similar to GPT-4 Code Interpreter*.

In this paper, leveraging the strengths of GPT-4 Code Interpreter (Zhou et al., 2023a), we introduce a simple yet effective framework, **MathCoder**, designed to enhance the mathematical reasoning capabilities of open-source models. This framework can be categorized into two parts: (1) math instruction-following dataset construction and (2) customized supervised fine-tuning. *Specifically*, the instruction-following dataset, termed as *MathCodeInstruct*, consists exclusively of 80k math problems and their corresponding solutions. Each solution is interwoven with *natural language for reasoning*, *code for execution*, and *execution results*. The comparison between `MathCodeInstruct` and other math instruction-tuning datasets is shown in Tab. 1.

`MathCodeInstruct` is created in two steps. The first step is collecting GPT-4 Code Interpreter-style solutions for the GSM8K and MATH training sets. GSM8K and MATH are two important datasets of math problems for improving and evaluating models' mathematical abilities, which consist of grade school math word problems and challenging competition mathematics problems, respectively. Using this data, we trained our initial models, termed *MathCoder-Initial*. The second step is to augment more math problems by using an innovative prompt named *problem interpolation*, which asks the LLM to generate questions with difficulty levels that fall between the provided MATH and GSM8K problems. This paradigm generates problems that bridge the gap between the grade-school-level problems in GSM8K and the challenging high-school-level problems in MATH, thus enhancing the dataset's generalization capability. We use MathCoder-Initial to generate solutions for these new problems. Combining this new data with those from the first step, we fine-tune the base Llama-2 models, reaching a score that outperforms the SOTA by a clear margin on GSM8K and MATH. Concurrently with our work, MAmmoTH (Yue et al., 2023) also creates a dataset consisting of math problems and model-generated solutions. However, their solutions consist of either only code or only natural language reasoning steps, which is notably different from our dataset of GPT-4 Code Interpreter-style solutions.

Regarding the supervised fine-tuning stage, we propose an effective training and inference pipeline to ensure that our fine-tuned model can behave in a manner similar to the GPT-4 Code Interpreter. We use special tokens (<|text|>, <|code|>, <|execution|>) to identify if a part of the training data is natural language, code, or execution results. With this deliberately created training corpus, the model learns to generate interleaved natural language and code divided by special tokens. During inference, we can use the special tokens to detect code blocks and utilize Jupyter Notebooks for code execution. We append the result of on-the-fly execution to the previous predictions of the model. Then, the model continues to autoregressively predict the next token based on this new version of the input, which includes the execution result at the end. In this way, the model would be able to "see" the execution results and continue its reasoning accordingly.

We use `MathCodeInstruct` to fine-tune popular open-source Llama-2 and CodeLlama (Rozière et al., 2023) models, creating a family of models named MathCoder. Experimental results show that

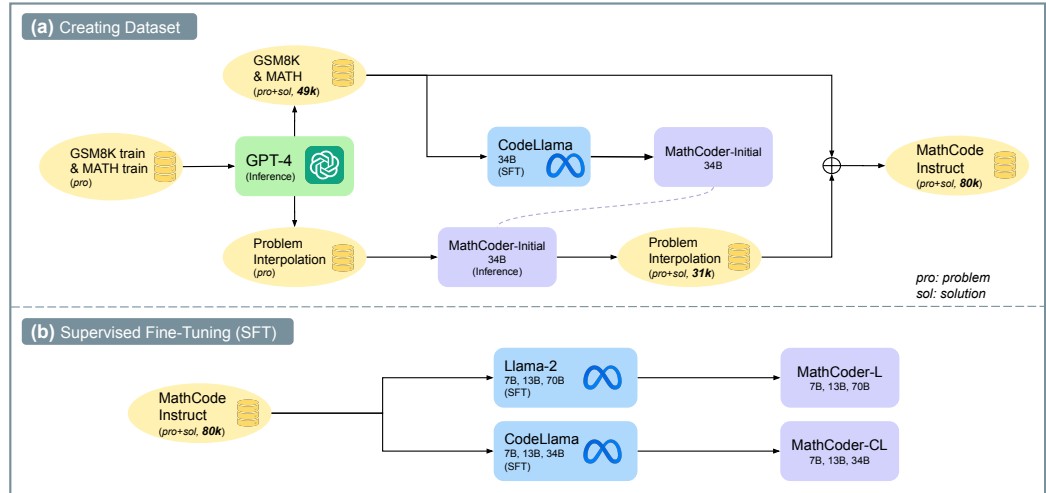

**Figure 2:** The process of dataset creation and model fine-tuning. **(a)** First, solutions for problems in the GSM8K and MATH datasets are collected from the GPT-4. Then, we fine-tune the CodeLlama-34B model on this data, producing the MathCoder-Initial. New problems are created using our novel prompt (detailed examples in Appendix C), and their solutions are generated using MathCoder-Initial. **(b)** Finally, the new problems and solutions are combined with the existing training data to create the final dataset, which we use to fine-tune the base Llama-2 model, producing our final MathCoder model.

the models with our proposed dataset and training framework achieve significant improvement on various mathematical reasoning benchmarks, as depicted in Fig. 1.

This paper's main contributions can be summarized in three key aspects:

- To the best of our knowledge, this is the first systematic study that explicitly integrates natural language reasoning, code generation, and feedback from execution results into open-source pre-trained large language models, aiming at enhancing their mathematical reasoning abilities.

- We have constructed a high-quality mathematical instruction tuning dataset, `MathCodeInstruct`. This dataset comprises existing math problems from GSM8K and MATH, with GPT-4 Code Interpreter-style solutions, and newly formulated ones via our novel problem interpolation prompting strategy.

- We have produced a family of models, MathCoder. We fine-tune Llama-2 and CodeLlama models on our dataset, producing a family of models with not only high accuracy on the GSM8K and MATH, but also a good performance on other out-of-domain datasets like Mathematics and SimulEq.

## 2 MATHCODER: SPECIALIZING LLAMA FOR MATHEMATICAL REASONING

In this section, we first introduce the methodology on creating `MathCodeInstruct` in Sec. 2.1. Subsequently, we detail the supervised fine-tuning (SFT) and inference methods in Sec. 2.2.

### 2.1 MATHCODEINSTRUCT DATASET

Our `MathCodeInstruct` dataset can be expressed as $D = \{D_0, D_1\}$, where $D_0$ denotes the seed data and $D_1$ is the data generated with the proposed prompting method, named *problem interpolation prompting*. Fig. 2 (a) outlines the process for creating `MathCodeInstruct` Dataset.

**Seed data $D_0$.** First, we obtain solutions for the GSM8K and MATH training sets from the GPT-4. The data can be expressed in (solution, question) pairs as $\{(y_i, x_i)\}_{i=1}^N$. Each solution $y_i$ contains three kinds of components: *natural language (text) for reasoning* **L**, *code for execution* **C**, and

---

**Input for MathCoder**

<|system|><|text|>Below is a math problem. Please solve it step by step.<|endofblock|><|endofmessage|>

<|user|><|text|>A group of friends went on a road trip across two cities. In the first city, they spent half of the money they had plus an additional \$50. In the second city, they spent half of what was left plus an additional \$20, leaving them with \$40. How much money did they start with before the road trip?<|endofblock|><|endofmessage|>

**(a) CoT example**

Let's call the amount of money they started with $x$.

1. In the first city, they spent half ($\frac{x}{2}$) plus \$50. What remains is $x - \left(\frac{x}{2} + 50\right)$.

2. Simplifying this gives $x - \frac{x}{2} - 50 = \frac{x}{2} - 50$.

3. In the second city, they spent half of what remained ($\frac{\frac{x}{2} - 50}{2}$) plus \$20. What remains is $\frac{x}{2} - 50 - \left(\frac{\frac{x}{2} - 50}{2} + 20\right)$.

4. Simplifying this gives $\frac{x}{2} - 50 - \frac{x}{4} + 25 - 20 = \frac{x}{4} - 45$.

5. They are left with \$40, so $\frac{x}{4} - 45 = 40$.

6. Solving for $x$ gives $\frac{x}{4} = 85$ and $x = 340$.

**(b) PoT example**

```
# Define a variable to represent the starting amount of money
starting_amount = 0
# Loop to find the starting amount
while True:
    amount = starting_amount
    # Spent in the first city: half of the money they had plus $50
    amount -= (amount / 2 + 50)
    # Spent in the second city: half of what was left plus $20
    amount -= (amount / 2 + 20)
    # Check if they are left with $40
    if amount == 40:
        break
    starting_amount += 1

starting_amount >>> 340
```

**(c) LCE solution from MathCoder**

<|assistant|> <|text|> To solve this problem, let's set $x$ as the amount of money the group of friends started with.
1. In the first city, they spent half of what they had ($\frac{x}{2}$) plus an additional \$50.
2. After this, they are left with $x - \left(\frac{x}{2} + 50\right)$.
3. In the second city, they spend half of what's left plus an additional \$20.
4. Finally, they are left with \$40.
The equation can be set up as follows:

$$x - \left(\frac{x}{2} + 50\right) - \left(\frac{x - \left(\frac{x}{2} + 50\right)}{2} + 20\right) = 40$$

Let's go ahead and solve this equation using code.<|endofblock|>

<|code|> from sympy import symbols, Eq, solve

```
# Define the variable and equation
x = symbols('x')

# Equation based on the problem description
equation = Eq(x - (x/2 + 50) - ((x - (x/2 + 50)) / 2 + 20), 40)

# Solve the equation
initial_amount = solve(equation, x)
initial_amount[0]
```
<|endofblock|>

<|execution|> 340<|endofblock|>

<|text|> Thus, the group of friends started with \$340 before the road trip.<|endofblock|><|endofmessage|>

**Figure 3:** Example of CoT (Wei et al., 2022), PoT (Gao et al., 2023; Chen et al., 2022) and LCE solution with special token. In contrast to CoT, which consists solely of natural language, and PoT, which consists solely of code, our LCE solution intertwines natural language, code, and execution results. <|text|>, <|code|>, and <|execution|> are special tokens that denote natural language, code, and execution results respectively.

*execution results* $\mathbf{E}$, where $\mathbf{L}$ is the natural language reasoning step, $\mathbf{C}$ is the Python code the model generates when its reasoning leads to some complex computation that it needs code to solve, and $\mathbf{E}$ is the output of the code. $\mathbf{E}$ is assessed by the model so a new $\mathbf{L}$ can be generated. All three kinds of components are closely chained together in the solutions, with each component influencing the component that comes after. An integral solution $y_i$ can be expressed as ($\mathbf{L}, \mathbf{C}, \mathbf{E}, \mathbf{L}, \mathbf{C}, \mathbf{E}, ...$). An example is shown in Fig. 3 (c). We call solutions in this format **Natural Language, Code, and Execution (LCE)** solutions. We put some case studies in Appendix H to demonstrate the advantage of LCE.

We filter the seed data $D_0 = (\{(y_i, x_i)\})$, making sure that each solution $y_i$ provides the same answer as the ground truth answer so that the quality of the dataset is further assured. Then, we fine-tune the CodeLlama-34B using the seed data $D_0$, producing our initial MathCoder model, named MathCoder-Initial.

**Problem interpolation prompting** $D_1$. Using the initial MathCoder model, we can generate LCE solutions for new problems. We observed a large gap in difficulty between grade-school-level GSM8K problems and challenging competition MATH problems. To bridge this gap, we present a novel prompting method (see details in Appendix C), which provides a powerful LLM like GPT-4 with a relatively simple problem drawn from the GSM8K training set, paired with a difficult problem drawn from the MATH, and ask the model to generate a new problem with difficulty between the two. GPT-4 generated completely novel intermediate-level problems, instead of just copying the problems from GSM8k and MATH. We then use GPT-4 to evaluate the new problems, and the results are shown in Fig. 4. We can see that 83.2% of the new problems are more difficult than

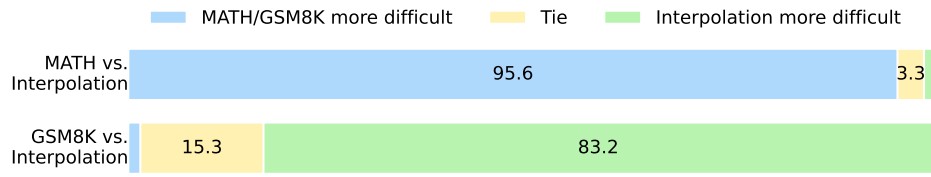

**Figure 4:** Difficulty comparison of interpolation problems against MATH and GSM8K using GPT-4. The evaluation prompt and examples are shown in Appendix E.

GSM8K, and 95.6% are easier than MATH, indicating that the problems generated in this way are appropriate in difficulty.

We also investigated using only GSM8K to create difficult problems, but we found that the new problems were too similar to the original ones, and the large gap to MATH still exists (more information can be found in Appendix F).

**Self-distillation.** We primarily use self-distillation due to the high cost of using GPT-4. As we observed that our MathCoder-Initial can already generate well-structured LCE-format solutions, we generated the solutions of $D_1$ with MathCoder-Initial. This does not affect the experiment's ability to assess the efficacy of problem interpolation prompting, because if solutions generated by a model weaker than GPT-4 can improve performance, then using GPT-4 might yield even greater improvements but leading to much more financial cost. Further discussion can be found in Appendix D. Given that we do not have ground truth answers for the new problems, we then generate $n$ different LCE solutions as depicted in (Wang et al., 2023a) for each new problem with our initial MathCoder models, keeping only those solutions for which all $n$ answers match ($n$ is set to 3 in this paper), thus ensuring our dataset's quality.

Combining the new data $D_1$ with the seed data $D_0$ yields the `MathCodeInstruct` dataset $D = \{D_0, D_1\}$. We fine-tune the base Llama-2 (Touvron et al., 2023) and CodeLlama (Rozière et al., 2023) models using `MathCodeInstruct` to derive our final MathCoder models. For clarity, we refer to the supervised fine-tuning of base Llama-2 as "MathCoder-L" and that of CodeLlama as "MathCoder-CL", as shown in Fig. 2 (b).

## 2.2 SUPERVISED FINE-TUNING AND INFERENCE

**Supervised Fine-tuning.** In order to identify the three kinds of components in LCE solutions, as illustrated in Fig. 3 (c), we enclose them with special tokens. *Reasoning language* starts with <|text|>, while *math code* and *execution results* start with <|code|> and <|execution|> respectively. All components end with <|endofblock|>. These tokens help the model understand the difference between each component and create LCE solutions during inference. After the special tokens are added, all components are concatenated to form the solution, which is preceded by the original math question to form an instance of training data. In order to make the training more efficient, several instances are concatenated together to form a single input, while cross-question masking is used to ensure only tokens in the same instance are visible.

During supervised fine-tuning, we apply a standard cross-entropy loss following Alpaca (Taori et al., 2023). The loss is only computed on *reasoning language* and *math code* since they are the components of the training data generated by the LLM. In particular, we *zero-out* the loss on tokens from *execution results*, as the model would not need to predict these tokens.

**Inference.** After supervised fine-tuning, the model has learned to output *natural language* and *code* enclosed by special tokens. We can identify the end of each component by looking for <|endofblock|>, and determine which component it is by examining the first token of the component. When a *code generation* is encountered, we utilize a Jupyter Notebook for real-time code execution, allowing the variables defined in previous code blocks to be used in subsequent ones. After execution, the execution results are concatenated following the previous *math code* block. The model then continues to autoregressively generate the next *reasoning language* block, forming the chain of thoughts in the LCE format, until it reaches the final answer. This process ensures that the model behaves similarly to the GPT-4 Code Interpreter.

## 3 EXPERIMENTS

### 3.1 DATASETS AND IMPLEMENTATION DETAILS

**Datasets.** We evaluate the MathCoder on five datasets, including two in-domain datasets: GSM8K (Cobbe et al., 2021) and MATH (Hendrycks et al., 2021); and three out-of-domain datasets: SVAMP (Patel et al., 2021), Mathematics (Saxton et al., 2019), and SimulEq (Kushman et al., 2014). We regard GSM8K and MATH as in-domain because their training sets are used for our supervised fine-tuning, while SVAMP, Mathematics, and SimulEq are out-of-domain because their training sets are not used in our fine-tuning. The extensive assortment of assessment datasets encompasses mathematical challenges from elementary, high school, and collegiate levels, covering various subjects like geometry, formal logic, and even commonsense reasoning. The selection of these datasets aims at providing a thorough evaluation of the models' ability to generalize to unknown circumstances and diverse fields of mathematics.

**Implementation Details.** Different base LLMs of varying sizes are tested, including Llama-2 (7B, 13B, and 70B) and CodeLlama (7B, 13B, and 34B). During training, we use a uniform learning rate of $2 \times 10^{-5}$ and a context length of 2048, and we set the batch size as 128 with different ratios of gradient accumulation steps and per-device train batch size, considering the model size. Additionally, we used a cosine scheduler for three epochs in total with a 50-step warm-up period. To efficiently train the computationally intensive models, we simultaneously employ DeepSpeed training with ZeRO-3 stage (Rajbhandari et al., 2020) and flash attention (Dao et al., 2022). The 7B, 13B, and 34B/70B models are trained on 8, 16, and 32 NVIDIA A800 80GB GPUs, respectively. The text-generation-inference framework of Hugging Face is used for inference with greedy decoding and max new tokens of every block set 512, and one to four GPUs are used as needed. We allow up to 32 LCE blocks in every solution.

**Baselines.** We compare the proposed MathCoders with the following competitive baselines. Closed-Source Models: we consider three closed-source models, including ChatGPT-3.5 Brown et al. (2020), GPT-4 (OpenAI, 2023), GPT-4 Code Interpreter (Zhou et al., 2023a), and PaLM-2 (Anil et al., 2023). Open-Source Models: we compare with Llama-2 (Touvron et al., 2023), WizardMath (Luo et al., 2023), Llama-1 RFT (Yuan et al., 2023), and Galactica (Taylor et al., 2022).

For baselines, CoT prompting (Wei et al., 2022) and few-shot in-context-learning (Dong et al., 2023) are used to maximize their performance while our MathCoders are always evaluated without extra prompting and under zero-shot setting (Kojima et al., 2023).

### 3.2 MAIN RESULTS

**Comparison between MathCoder and SOTA open-source models.** The experiment results in Tab. 2 show that our method outperforms other open-source competitive math-solving models with a clear advantage, achieving state-of-the-art results across all datasets. However, a substantial performance gap still exists compared to the state-of-the-art closed-source method GPT-4 Code Interpreter. Our observations are as follows: (1) *MathCoder-L-7B outperforms WizardMath-70B.* Even the smallest version of MathCoder, MathCoder-L-7B, outperforms the largest WizardMath model, WizardMath-70B, on three out of five datasets, achieving a significant gain (+4.5%) in the average score, as shown in Tab. 2. This is likely attributed to the fact that WizardMath is trained solely on CoT data, while MathCoder is trained on our proposed LCE solutions. This demonstrates the advantage of using solutions that interleave natural language, code, and execution (LCE blocks), significantly enhancing the model's ability to perform complex computations. (2) Additionally, it is worth noting that while the code ability of CodeLlama-34B significantly outperforms that of Llama-2-70B, in the case of MathCoder models, we observed that models based on Llama-2-70B (73.1%) can outperform CodeLlama-34B (70.2%). This contrasts with the findings in the concurrent work, MAmmoTH (Yue et al., 2023). The main reason for this disparity might be that Llama-2-70B exhibits better natural language reasoning ability, and the `MathCodeInstruct` dataset can enhance language models' code generation ability for math problem-solving.

**Comparison between Llama-2 and CodeLlama.** Tab. 3 shows that MathCoder-CL with CodeLlama as the base model brings a substantial improvement compared to MathCoder-L with Llama-2 as the base model. MathCode-CL-7B and MathCoder-CL-13B demonstrate an accuracy improvement of 4.1% and 3.0% respectively, compared to the corresponding MathCoder-L models

**Table 2:** Model evaluation on in-domain (GSM8K & MATH) and out-of-domain datasets (SVAMP, Mathematics & SimulEq). + incidates improvement w.r.t. the best open source model. SVA. stands for SVAMP, Mat. stands for Mathematics, and Sim. stands for SimulEq.

| Model | Base | Size | In-Domain | | Out-of-Domain | | | Average |
|---|---|---|---|---|---|---|---|---|
| | | | GSM8K | MATH | SVA. | Mat. | Sim. | |
| **Closed-Source Model** | | | | | | | | |
| ChatGPT-3.5 (Zhao et al., 2023) | - | - | 80.8 | 34.1 | - | - | - | - |
| GPT-4 Code (Zhou et al., 2023a) | - | - | 97.0 | 69.7 | - | - | - | - |
| PaLM-2 (Anil et al., 2023) | - | - | 80.7 | 34.3 | - | - | - | - |
| **Open-Source Model** | | | | | | | | |
| Llama-1 RFT (Yuan et al., 2023) | Llama-1 | 34B | 56.5 | 7.4 | 55.4 | 7.6 | 12.8 | 27.9 |
| WizardMath (Luo et al., 2023) | Llama-2 | 7B | 54.9 | 10.7 | 36.1 | 9.3 | 12.8 | 24.8 |
| | | 13B | 63.9 | 14.0 | 51.9 | 14.1 | 14.9 | 31.8 |
| | | 70B | 81.6 | 22.7 | 71.8 | 17.1 | 37.9 | 46.2 |
| MathCoder-L | Llama-2 | 7B | 64.2 | 23.3 | 71.5 | 46.9 | 47.5 | 50.7 |
| | | | +9.3 | +12.6 | +35.4 | +37.6 | +34.7 | +25.9 |
| | | 13B | 72.6 | 29.9 | 76.9 | 54.7 | 62.3 | 59.2 |
| | | | +8.7 | +15.9 | +25.0 | +40.6 | +47.4 | +27.4 |
| | | 70B | 83.9 | 45.1 | 84.9 | 74.4 | 77.0 | 73.1 |
| | | | +2.3 | +22.4 | +13.1 | +57.3 | +39.1 | +26.9 |
| MathCoder-CL | CodeLlama | 7B | 67.8 | 30.2 | 70.7 | 55.8 | 49.6 | 54.8 |
| | | | +12.9 | +19.5 | +34.6 | +46.5 | +36.8 | +30.0 |
| | | 13B | 74.1 | 35.9 | 78.0 | 62.5 | 60.7 | 62.2 |
| | | | +10.2 | +21.9 | +26.1 | +48.4 | +45.8 | +30.4 |
| | | 34B | 81.7 | 45.2 | 82.5 | 75.9 | 65.8 | 70.2 |
| | | | +0.1 | +22.5 | +10.7 | +58.8 | +27.9 | +24.0 |

**Table 3:** Model performance comparison for MathCoders with CodeLlama and Llama-2 as base.

| Size | GSM8K | MATH | SVAMP | Mathematics | SimulEq | Average |
|---|---|---|---|---|---|---|
| MathCoder-**C**L-7B vs. MathCoder-L-7B | +3.6 | +6.9 | -0.8 | +8.9 | +2.1 | +4.1 |
| MathCoder-**C**L-13B vs. MathCoder-L-13B | +1.5 | +6.0 | +1.1 | +7.8 | -1.6 | +3.0 |

of the same size. The potentially superior coding and reasoning capability of CodeLlama can be attributed to its additional training on code data (Rozière et al., 2023). This extended training provides CodeLlama with a deeper understanding of programming concepts and patterns, allowing it to excel in coding-related tasks and exhibit more advanced math reasoning abilities.

**Comparison among different subjects across various levels.** MATH dataset problems are categorized with difficulty levels ranging from 1 to 5, covering seven different math subjects, including algebra, prealgebra, number theory, counting and probability, precalculus, intermediate algebra, and geometry. In Fig. 5, we present the performance comparison of MathCoder-L (7B, 13B) and MathCoder-CL (7B, 13B), grouped by these levels and subjects. More results are shown in Appendix G. We find that MathCoder achieves higher scores in algebra and prealgebra problems. However, when it comes to geometry problems, MathCoder struggles to achieve high scores, especially for problems with higher difficulty levels. This suggests that code plays a more significant role in computationally intensive questions.

## 3.3 ABLATION STUDY

**Analysis of the influence of problem interpolation.** We conducted an experiment to study the influence of the portion of `MathCodeInstruct` questions created using the proposed problem interpolation. The experiment uses CodeLlama-34B as the base model. The experimental results in Tab. 4 validate that problem interpolation brings a significant improvement across all five datasets. We also conducted an experiment where we generated 31k data samples using GSM8K or MATH as the seed data separately with equal portions. The results are presented in Tab. 8. As shown, experiments involving problem interpolation yield an average accuracy that is 3.6 percentage points

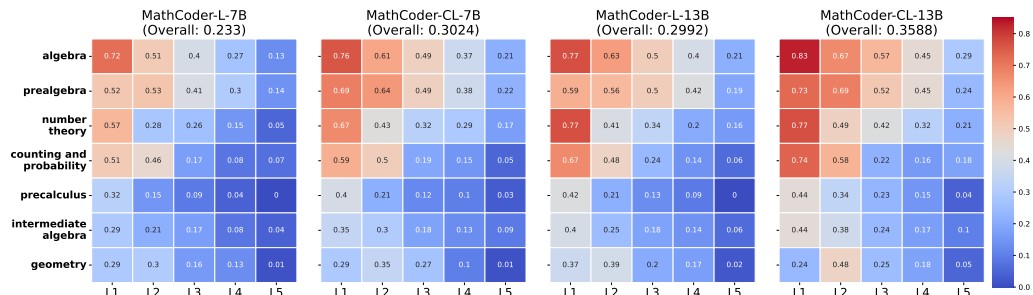

**Figure 5:** Performance comparison of MathCoder-L (7B, 13B) and MathCoder-CL (7B, 13B) on the MATH dataset by levels and subjects. We can see that the improved accuracy from MathCoder-L to MathCoder-CL comes primarily from subjects that require precise calculations like algebra and number theory.

**Table 4:** Influence of the interpolation problems in `MathCodeInstruct` (as shown in Tab. 1) based on CodeLlama-34B.

| Train set | Samples | GSM8K | MATH | SVAMP | Mathematics | SimulEq | Average |
|---|---|---|---|---|---|---|---|
| GSM8K+MATH | 49k | 77.3 | 44.0 | 78.6 | 71.6 | 59.3 | 66.2 |
| GSM8K+MATH+Interpolation | 80k | 81.7 **+4.4** | 45.2 **+1.2** | 82.5 **+3.9** | 75.9 **+4.3** | 65.8 **+6.4** | 70.2 **+4.0** |

higher compared to those without it. These results indicate that by employing problem interpolation, we can generate problems with intermediate difficulty levels, thereby increasing the diversity of the problem set. This expands the diversity of the problems and ultimately enhances the performance of the model. Further experiments on the effects of different amounts of data created with problem interpolation are presented in Tab. 9 (Appendix B).

**Analysis of LCE solutions compared to code-only or natural-language-only solutions.** To analyze the advantages brought by the LCE solutions, consisting of interleaved natural language, code, and execution results, we trained a new model with solutions consisting of code-only. We use the results of WizardMath 7B Luo et al. (2023), which was trained on natural language, to represent the performance of natural-language-only solutions. The results are shown in Tab. 5 and Tab. 6. As can be seen, the LCE solution produces the highest average accuracy, surpassing the code-only solution by 17.9 percentage points and the natural-language-only solution by 25.9 percentage points.

**Analysis of Code Execution.** To demonstrate the effect of code execution, both in training time and execution time, we have done further experiments. The results and analysis are presented in Appendix A.

## 4 RELATED WORK

**Instruction Tuning.** Instruction tuning is a method of enhancing LLMs' instruction following abilities, thus aligning language models with more useful objectives and human preferences. A long line of previous works (Ye et al., 2021; Longpre et al., 2023; Sanh et al., 2021; Wang et al., 2022b; Wei et al., 2021; Chung et al., 2022; Longpre et al., 2023) is focused on enhancing LLMs' instruction following abilities in general. With the emergence of models like GPT-3 and GPT-4, recent studies (Wang et al., 2022a; 2023b; Zhou et al., 2023b; Peng et al., 2023; Xu et al., 2023) have started to utilize synthetic instructions generated by these powerful models to tune smaller models. Compared to these works, our instruction tuning is focused on using high-quality solutions for math problems generated by models to improve our LLM's math-solving ability. Another related work is presented in (Luo et al., 2023), but their method did not use code to solve math problems, distinguishing our work from theirs.

**Mathematical Reasoning.** There are various benchmark datasets (Hendrycks et al., 2020; Ling et al., 2017; Hendrycks et al., 2021) to measure a model's mathematical reasoning abilities. Recently, many works have focused on enhancing LLMs' ability to solve math problems, reaching high scores

**Table 5:** Comparison between LCE and code-only solutions. Results of LCE-format and Only Code and Execution are acquired from models trained based on CodeLlama-7B.

| Solution Format | GSM8K | MATH | SVAMP | Mathematics | SimulEq | Average |
|---|---|---|---|---|---|---|
| LCE-format (ours) | 67.8 | 30.2 | 70.7 | 55.8 | 49.6 | 54.8 |
| Only Code and Execution | 50.2 **-17.6** | 20.2 **-10.0** | 61.6 **-9.1** | 39.8 **-16.0** | 12.8 **-36.8** | 36.9 **-17.9** |

**Table 6:** Comparison between LCE and natural-language-only solutions. Both the LCE format model and WizardMath (Luo et al., 2023) are finetuned from Llama-2-7B.

| Solution Format | GSM8K | MATH | SVAMP | Mathematics | SimulEq | Average |
|---|---|---|---|---|---|---|
| LCE-format (ours) | 64.2 | 23.3 | 71.5 | 46.9 | 47.5 | 50.7 |
| Only Natural Language (WizardMath 7B) | 54.9 **-9.3** | 10.7 **-12.6** | 36.1 **-35.4** | 9.3 **-40.3** | 12.8 **-34.7** | 24.8 **-25.9** |

on these benchmarks. Many of them apply Chain-of-Thought (Wei et al., 2022; Kojima et al., 2023; Wang et al., 2023a; Fu et al., 2022) to improve LLMs' multistep reasoning capability. Another line of works (Gao et al., 2023; Chen et al., 2022; Zhou et al., 2023a) utilize code to compensate for LLMs' limitations in doing complex math computations. Our work takes inspiration from these two lines of work, as we believe both Chain-of-Thought and code generation (Li et al., 2023a; Rozière et al., 2023) are essential to solving math problems. There are also works focused on math-related pre-training (Lewkowycz et al., 2022; Taylor et al., 2022) to improve a model's general reasoning capability. We combine natural language and code seamlessly in our dataset, thus providing a method to train models more efficiently in solving math problems.

**Distillation.** Distillation (Hinton et al., 2015) often involves transferring knowledge from a larger, more powerful model to a smaller, weaker one (Taori et al., 2023; Zheng et al., 2023; Cobbe et al., 2021). Recent research (Li et al., 2023b; Wang et al., 2022a; Allen-Zhu & Li, 2020) has demonstrated the plausibility of self-distillation, achieving performance improvements by distilling the model itself. Our approach can also be viewed as a form of self-distillation, as the solutions generated by MathCoder-Initial, which is built on CodeLlama-34B, are used to fine-tune CodeLlama-34B, resulting in MathCoder-CL-34B.

## 5 CONCLUSION AND LIMITATION

In this paper, we present MathCoder, an open-source large language model designed for math reasoning, bridging the gap between natural language understanding and computational problem-solving. MathCoder incorporates math instruction-following dataset construction. By utilizing the GSM8K and MATH datasets as seed data, we leverage the GPT-4 to generate problems encompassing reasoning, code generation, and program execution. Additionally, we propose a problem interpretation method to create intermediate-level problems. Furthermore, we introduce a customized supervised fine-tuning approach, where the training loss is only applied to natural language and code. Our empirical study demonstrates that MathCoder achieves state-of-the-art performance in five math datasets among open-source LLMs, with scores of 83.9% on the GSM8K dataset and 45.2% on the MATH dataset. It is worth noting that MathCoder outperforms closed-source models like ChatGPT-3.5 and PaLM-2 on the GSM8K and MATH datasets and even outperforms GPT-4 on the MATH dataset.

However, our work does have certain limitations that warrant further exploration in future research. First, since we rely on the GPT-4 for data generation, MathCoder's capabilities are inherently constrained by the capabilities of this model and unable to solve theorem-proving problems. Additionally, as a series of uni-modal models, MathCoder still faces challenges in solving complex geometry problems, which we acknowledge and plan to address in our future investigations.

# 6 ACKNOWLEDGMENTS

This project is funded in part by National Key R&D Program of China Project 2022ZD0161100, and in part by General Research Fund of Hong Kong RGC Project 14204021.

# 7 AUTHOR CONTRIBUTION STATEMENT

Hongsheng Li and Mingjie Zhan led the project. Ke Wang, Aojun Zhou and Zimu Lu were responsible for proposing the methodology, conducting experiments, and contributing to the manuscript. Houxing Ren developed the codebase for code generation and offered suggestions for its improvement. Weikang Shi, and Sichun Luo assisted with manuscript writing and actively participated in discussions. Additionally, Renrui Zhang and Linqi Song also contributed to these discussions.

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

**Table 7:** Ablation study of with/without code execution during *inference* and of the loss with/without execution results in *training* stage.

| Experiment | Include execution results for training | Actual code execution in inference | GSM8K | MATH | SVAMP | Mathematics | Simuleq | Average |
|---|---|---|---|---|---|---|---|---|
| #1 | Yes | No | 54.1 | 16.9 | 69.6 | 20.6 | 14.2 | 35.1 |
| #2 | Yes | Yes | 79.9 **+25.8** | 45.9 **+29.0** | 81.9 **+12.3** | 74.2 **+53.6** | 63.6 **+49.4** | 69.1 **+34.0** |
| #3 | No | Yes | 81.7 **+1.8** | 45.2 **-0.7** | 82.5 **+0.6** | 75.9 **+1.7** | 65.8 **+2.1** | 70.2 **+1.1** |

## A ANALYSIS IF CODE EXECUTION

**Analysis of actual code execution in the inference stage.** We investigate the impact of code execution in the inference stage and report the results in Tab. 7. We conduct this investigation using CodeLlama-34B as the base model and train the models on our 80k `MathCodeInstruct` dataset. Tab. 7 (#1) and Tab. 7 (#2) use the same model, trained with the cross-entropy loss computed on not only natural language and code, but also the execution results. In this way, this model learns to predict the execution results. In Tab. 7 (#1), the code execution results are predicted by the model itself, while in Tab. 7 (#2), the execution result is returned from a Python code interpreter. From the comparison between Tab. 7 (#1) and Tab. 7 (#2), we can see that Tab. 7 (#2) outperforms Tab. 7 (#1) across all five datasets, showing an improvement of 34.0% in the average accuracy score. This indicates that actual code execution in the inference stage has a significant impact on the model's performance. This study shows that the model failed to predict correct execution results for many programs and that actually executing the code using an external tool can significantly improve the accuracy while doing complex computations. This finding validates the significance of integrating code execution when solving math problems with LLMs, in line with previous closed-source GPT-4 Code Interpreter (Zhou et al., 2023a).

**Analysis of execution results in the training stage.** Based on the observation that actual code execution contributes a lot to the model's performance, we investigate not forcing the model to predict the correct execution result. Tab. 7 (#3) is the performance of MathCoder-CL-34B, which ignores execution results when computing the loss, so that the model does not learn to estimate the execution results and the learning task at the supervised fine-tuning stage becomes simpler. Compared to Tab. 7 (#2), Tab. 7 (#3) improves the accuracy across four out of five datasets, resulting in a rise in the average accuracy from 69.1% to 70.2%, which aligns with the hypothesis that by computing the loss only on natural language and code, the model can focus more on the math problem-solving skills itself, thus making the supervised fine-tuning more effective.

## B ADDITIONAL EXPERIMENTS

In this section, we present the results of additional ablation studies in Tab. 5, Tab. 6, Tab. 8, and Tab. 9.

### B.1 COMPARISON BETWEEN LCE FORMAT AND NATURAL-LANGUAGE-ONLY OR CODE-ONLY FORMAT

Tab. 5 and Tab. 6 compares our LCE solution with two other common solution formats: code-only and natural-language-only. LCE solution produces the highest average accuracy, surpassing code-only solution by 17.9%, and natural-language-only solution by 25.9%.

### B.2 ANALYSIS OF USING SINGLE DATASET AND USING PROBLEM INTERPOLATION

Tab. 8 presents a comparison between using problems from both the GSM8K and MATH datasets for problem interpolation, and using problems from only a single dataset for data augmentation. As can be seen, experiments with problem interpolation produce an average accuracy that is 3.6 percentage points higher than those without it.

## B.3 ANALYSIS OF USING DIFFERENT NUMBER OF PROBLEM INTERPOLATION SAMPLES

Tab. 9 demonstrates the impact of using different numbers of problem interpolation samples. As presented in the table, the average accuracy continues to rise as the number of problem interpolation samples increases.

**Table 8:** Ablation study of problem interpolation using CodeLlama-7B.

| Training Data | GSM8K | MATH | SVAMP | Mathematics | SimulEq | Average |
|---|---|---|---|---|---|---|
| w/ interporation | 67.8 | 30.2 | 70.7 | 55.8 | 49.6 | 54.8 |
| w/o interporation | 61.9 | 29.1 | 70.9 | 50.5 | 43.4 | 51.2 **(-3.6)** |

**Table 9:** Ablation study of different numbers of problem interpolation samples. 49k is the number of D0 data. 0, 11k, 31k, and 51k denote different numbers of problem interpolation samples.

| Data Size | GSM8K | MATH | SVAMP | Mathematics | SimulEq | Average |
|---|---|---|---|---|---|---|
| 49k | 50.6 | 22.9 | 53.2 | 46.0 | 29.6 | 40.5 |
| 49k+11k | 56.4 | 26.8 | 64.9 | 47.6 | 40.7 | 47.3 **(+6.8)** |
| 49k+31k | 67.8 | 30.2 | 70.7 | 55.8 | 49.6 | 54.8 **(+14.3)** |
| 49k+51k | 68.0 | 32.6 | 70.9 | 60.1 | 52.7 | 56.9 **(+16.4)** |

## C  DATASET EXAMPLES

In this part, we include two examples that show the process of creating `MathCodeInstruct`. Fig. 6 shows an example with only one LCE block, while Fig. 7 shows an example with three LCE blocks.

## D  SOLUTIONS OF PROBLEM INTERPOLATION SAMPLES GENERATED WITH GPT4

To validate that replacing data generated by MathCoder-Initial with solutions generated by GPT4 can further improve accuracy, we generated solutions using GPT4 with additional funding, trained the model, and presented the results in Tab. 10. As expected, employing GPT4-generated data led to even better performance.

## E  EXAMPLES OF DIFFICULTY COMPARISON

We show five examples of using GPT-4 to evaluate the complexity of problems in `MathCodeInstruct`. Fig. 8 and Fig. 9 are two examples that the newly generated interpolation problems are more difficult than the origin GSM8K problems, and Fig. 10 is an example that the origin MATH problem is more difficult than the newly generated interpolation problem. These two situations are the most common (83.2% and 95.6%).

Fig. 11 shows an example that the newly generated interpolation problem ties with the origin GSM8K problem, which situation accounts for 15.3% of all problems.

Fig. 12 shows an uncommon example that the origin GSM8K problem is slightly more difficult than the newly generated interpolation problem according to GPT-4, which situation accounts for less than 3% of all problems.

**Table 10:** Comparison between GPT4-generated data and MathCoder-Initial-generated data.

| Base Model | Data Composition | GSM8K | MATH | SVAMP | Mathematics | SimulEq | Average |
|---|---|---|---|---|---|---|---|
| CodeLlama 7B | 49k (GPT-4) + 31k (MathCoder-Initial) | 67.8 | 30.2 | 70.7 | 55.8 | 49.6 | 54.8 |
| CodeLlama 7B | 80k (GPT-4) | 68.4 | 31.2 | 76.3 | 61.6 | 52.5 | 58.0 (+3.2) |
| CodeLlama 34B | 49k (GPT-4) + 31k (MathCoder-Initial) | 81.7 | 45.2 | 82.5 | 75.9 | 65.8 | 70.2 |
| CodeLlama 34B | 80k (GPT-4) | 82.2 | 47.6 | 84.1 | 79.2 | 69.7 | 72.6 (+2.4) |

# F    CREATING PROBLEMS USING ONLY GSM8K

Fig. 13, Fig. 14, Fig. 15, Fig. 16 and Fig. 17 are five examples that utilize problems from the train set of GSM8K to generate new problems which are more difficult than the origin ones. Compared with the problems generated by our interpolation method, we can see that the new problems generated in this way are much more similar to the raw GSM8K problems, sometimes just changing the name of some variables or scaling the value. These problems are only slightly more complicated than the raw problems, if not equally difficult, and are still much simpler than those from the MATH dataset.

In contrast to using just GSM8K, introducing problems from the MATH dataset in the interpolation method shows the model (GPT-4 here) a route to generate more challenging problems. Hence, the newly generated problems are similar to the problems in the GSM8K and the problems in the MATH. Consequently, these interpolation problems can narrow the difficulty gap between the two datasets.

# G    MORE EXPERIMENT RESULTS

We show the performance comparison of all MathCoders, MathCoder-L (7B, 13B, 70B) and MathCoder-CL (7B, 13B, 34B), on the MATH dataset by levels and subjects in Fig. 18. We can see that the improved accuracy from MathCoder-L to MathCoder-CL comes primarily from subjects requiring precise calculations like algebra, number theory, and counting and probability.

# H    CASE STUDY WITH CoT, PoT AND LCE

We compare our LCE solutions with CoT solutions and PoT solutions. Fig. 19 is an example of a problem in number theory, and Fig. 20 is an example of a problem in algebra. CoT and PoT failed to solve the problem in both cases, but LCE succeeded.

Fig. 21, Fig. 22, and Fig. 23 are three solutions to one problem in geometry. The CoT solutions successfully figured out the coordinates of D, C, and E but failed to calculate the area, while the PoT could not interpret the conditions in the problem. Compared with them, we can see that our LCE solutions not only can utilize the conditions in the problems correctly but also will not make errors in calculation.

---

**(a) Interpolation Prompt:**

Please create a new problem, following the given example, "Example 1" is an easy problem and "Example 2" is much more difficulty than "Example 1", the new problem should be harder than "Example 1" and simpler than "Example 2".

Example 1: Marissa is hiking a 12-mile trail. She took 1 hour to walk the first 4 miles, then another hour to walk the next two miles. If she wants her average speed to be 4 miles per hour, what speed (in miles per hour) does she need to walk the remaining distance?

Example 2: The two solutions of equation $x^2 + bx + 48 = 0$ are in the ratio of 3 to 1 for some values of $b$. What is the largest possible value of $b$?

---

**(b) According new problem generated by GPT-4:**

A group of friends went on a road trip across three cities. In the first city, they spent half of the money they had plus an additional \$50. In the second city, they spent half of the remaining money plus an additional \$30. In the third city, they spent half of what was left plus an additional \$20, leaving them with \$40. How much money did they start with before the road trip?

---

**(c) Solution generated by MathCoder-Initial without extra prompt:**

Let's break this problem down step by step:
1. Let $x$ be the amount of money they started with.
2. After visiting the first city, they spent $\frac{x}{2} + 50$. So, the remaining amount is $x - \left(\frac{x}{2} + 50\right)$.
3. In the second city, they spent half of the remaining money plus \$30. This means they spent $\frac{x-\left(\frac{x}{2}+50\right)}{2} + 30$. After this expenditure, they would have $x - \left(\frac{x}{2} + 50\right) - \left(\frac{x-\left(\frac{x}{2}+50\right)}{2} + 30\right)$ left.

4. In the third city, they spent half of what was left from the above amount plus \$20. After this, they were left with \$40.
Using the above information, we can set up an equation to solve for $x$:

$$x - \left(\frac{x}{2} + 50\right) - \left(\frac{x - \left(\frac{x}{2}+50\right)}{2} + 30\right) - \left(\frac{x - \left(\frac{x}{2}+50\right) - \left(\frac{x-\left(\frac{x}{2}+50\right)}{2} + 30\right)}{2} + 20\right) = 40$$

Let's solve this equation to find the value of $x$.

```python
from sympy import symbols, Eq, solve

# Define the variable
x = symbols('x')

# Define the equation based on the problem statement
equation = Eq(x - (x/2 + 50) - (x - (x/2 + 50))/2 - 30 - (x - (x/2 + 50) - (x - (x/2 + 50))/2
    - 30)/2 - 20, 40)

# Solve the equation
initial_money = solve(equation)[0]

>>> 700
```

The group of friends started with \$700 before the road trip.

**Figure 6:** An example of the process of creating `MathCodeInstruct`. Firstly, "Example 1" and "Example 2" are randomly chosen from the train set of GSM8K and MATH respectively. Then a new problem is generated by GPT-4 using the interpolation prompt. Finally, we use our initial MathCoder to generate LCE-style solution for the new problem.

**(a) Interpolation Prompt:**

Please create a new problem, following the given example, "Example 1" is an easy problem and "Example 2" is much more difficulty than "Example 1", the new problem should be harder than "Example 1" and simpler than "Example 2".

Example 1: A bus comes by Jerry's bus stop every 20 minutes starting at exactly 5:13 a.m. If Jerry shows up at exactly 8:35 a.m., how many minutes will he have to wait for the next bus?

Example 2: Bella's grandfather is confined to a wheelchair. He is coming to visit her. Bella wants to build a wheelchair ramp. Her research shows that there must be 3.5 meters of ramp for every 30 centimeters of elevation. The distance from the ground to the front doorstep of Bella's house is 9 cm. What must the length of the ramp be?

**(b) According new problem generated by GPT-4:**

Liam is a bird watcher and records the pattern of a particular species of bird migration. Every 15 days, the birds fly south 250 miles and then fly back north 100 miles. If this pattern continues, how far south from their original position will the birds be after 75 days?

**(c) Solution generated by MathCoder-Initial without extra prompt:**

Let's solve the problem step by step:
1. Calculate the net distance the birds have moved from their original position in 15 days.
2. Use the net distance to determine how far the birds will have moved after 75 days.
Let's start by calculating the net distance the birds fly south in 15 days. The net distance can be found by subtracting the northward distance from the southward distance.

```
south_distance_15_days, north_distance_15_days = 250, 100
net_distance_15_days = south_distance_15_days - north_distance_15_days
```

$>>>150$

The birds have moved a net distance of 150 miles south from their original position in 15 days.
Now, to determine how far the birds will have moved after 75 days, we need to:
Calculate how many 15-day cycles are within 75 days. Multiply the number of 15-day cycles by the net distance moved in 15 days to get the total distance moved in 75 days. Let's find out how many 15-day cycles are within 75 days.

```
total_days = 75
number_of_cycles = total_days // 15
```

$>>>5$

There are 5 cycles of 15 days within the span of 75 days.
Now, let's calculate the total distance the birds will have moved after 75 days by multiplying the number of 15-day cycles by the net distance moved in one 15-day cycle.

```
total_distance_75_days = number_of_cycles*net_distance_15_days
```

$>>>750$

After 75 days, the birds will be 750 miles south from their original position.

**Figure 7:** An example of the process of creating `MathCodeInstruct`. Firstly, "Example 1" and "Example 2" are randomly chosen from the train set of GSM8K and MATH respectively. Then a new problem is generated by GPT-4 using the interpolation prompt. Finally, we use our initial MathCoder to generate LCE-style solution for the new problem.

**(a) Evaluation Prompt:**

Which problem is more difficult?

Problem 1: "Zain has 10 more of each coin than Emerie. If Emerie has six quarters, seven dimes, and five nickels, how many coins does Zain have?"

Problem 2: "In a marathon, every runner must run through several checkpoints. In the first segment, Angela runs 15 meters more than Miguel. For each subsequent segment, Angela runs 5 meters less than she did in the previous segment, while Miguel runs consistently. If Miguel runs 50 meters in the first segment, how many meters does Angela run in total when she completes 5 segments?"

You answer should be one of "Problem 1", "Problem 2" and "Tie".

**(b) Judgement from GPT-4:**

Let's break down both problems step by step to assess their complexity.

**Problem 1:**
Emerie has:

- Quarters: 6
- Dimes: 7
- Nickels: 5

Zain has 10 more of each coin than Emerie:

- Quarters: $6 + 10 = 16$
- Dimes: $7 + 10 = 17$
- Nickels: $5 + 10 = 15$

To find out how many coins Zain has, we just need to add these numbers together:

$$16 + 17 + 15 = 48 \text{ coins}$$

**Problem 2:**
Given:
- Miguel runs consistently at 50 meters per segment.
- Angela runs $50 + 15 = 65$ meters in the first segment.
- Angela runs 5 meters less in each subsequent segment.
Using the concept of arithmetic series:

$$S_n = \frac{n}{2}(2a_1 + (n-1)d)$$

For Angela, the first term, $a_1$, is 65 meters (15 meters more than Miguel's 50 meters), and the common difference, $d$, is -5 meters, as she runs 5 meters less in each subsequent segment. She completes 5 segments, so $n = 5$.
Let's plug in the values to find out how many meters Angela runs in total for the 5 segments:

$$S_5 = \frac{5}{2}(2 \cdot 65 + (5-1) \cdot -5) = \frac{5}{2}(130 - 20) = \frac{5}{2}(110) = \frac{550}{2} = 275 \text{ meters}$$

**Conclusion:**
Problem 1 involves simple arithmetic and requires the solver to add a constant number of coins to the known number of coins that Emerie has. There's no series or sequential decrease/increase involved. Problem 2, however, involves understanding an arithmetic progression. It needs understanding of sequences and series, and it may require more steps to solve, such as identifying the first term, the common difference, and then applying the formula to find the sum of the series. Therefore, based on these considerations, it could be argued that "Problem 2" is more difficult.

**Figure 8:** An example of using GPT-4 to evaluate the complexity of problems in `MathCodeInstruct`. "Problem 2" is in `MathCodeInstruct` and "Problem 1" is the problem from GSM8K that was used to generate "Problem 2".

**(a) Evaluation Prompt:**

Which problem is more difficult?

Problem 1: "Peter has 15 birds. 1/3 are ducks. The rest are chickens and require special feed that costs $2 per bird. How much does it cost to feed the chickens?"

Problem 2: "In a rectangular garden, the length is 4 times the width. If the total area of the garden is increased by 120 square feet by increasing the width by 5 feet while keeping the length constant, what is the original area of the garden?"

You answer should be one of "Problem 1", "Problem 2", and "Tie".

**(b) Judgement from GPT-4:**

Let's break down the two problems to understand the complexity of each.

**Problem 1:**
Peter has 15 birds, and 1/3 are ducks. So the number of chickens is $\frac{2}{3} \times 15 = 10$ chickens.

The cost to feed the chickens is $2 per chicken, so:

$$\text{Cost} = 10 \text{ chickens} \times \$2/\text{chicken} = \$20$$

**Problem 2:**
Let the original width of the garden be $w$ and the length be $4w$.
The original area of the garden is $4w^2$.
If the width is increased by 5 feet, the new width is $w + 5$.
The new area is then $4w(w + 5) = 4w^2 + 20w$.
Given the total area of the garden is increased by 120 square feet:

$$4w^2 + 20w - (4w^2) = 120$$
$$20w = 120$$
$$w = 6 \text{ feet}$$

So, the original area is:

$$4w^2 = 4(6)^2 = 144 \text{ square feet}$$

**Comparison:**
Problem 1 involves basic arithmetic and a simple fraction, making it relatively straightforward. Problem 2 involves solving a quadratic equation and requires a better understanding of algebraic concepts and the area of rectangles.

Thus, **Problem 2** is more difficult than Problem 1.

**Figure 9:** An example of using GPT-4 to evaluate the complexity of problems in `MathCodeInstruct`. "Problem 2" is in `MathCodeInstruct` and "Problem 1" is the problem from GSM8K that was used to generate "Problem 2".

**(a) Evaluation Prompt:**

Which problem is more difficult?

Problem 1: "he two solutions of the equation $x^2 + bx + 48 = 0$ are in the ratio of 3 to 1 for some values of $b$. What is the largest possible value of $b$?"

Problem 2: "In a rectangular garden, the length is 4 times the width. If the total area of the garden is increased by 120 square feet by increasing the width by 5 feet while keeping the length constant, what is the original area of the garden?"

You answer should be one of "Problem 1", "Problem 2" and "Tie".

**(b) Judgement from GPT-4:**

The two problems provided, at their core, involve algebraic manipulation, but Problem 1 involves quadratic equations and a bit more conceptual understanding of quadratic roots, whereas Problem 2 is more straightforward and involves linear equations. Based on this, it can be said that "Problem 1" is more difficult compared to "Problem 2". Here's a brief explanation and solution for both problems to illustrate this point:

**Problem 1:**
Given that the roots of $x^2 + bx + 48 = 0$ are in the ratio $3 : 1$, let the roots be $3k$ and $k$. The sum of the roots is $4k$, and the product of the roots is $48$.
Using Vieta's formulas:
- $3k + k = 4k = -b$
- $3k * k = 3k^2 = 48$
Solving for $k$ in the second equation:

$$k^2 = 16 \implies k = \pm 4$$

So, the roots are $12$ and $4$ or $-12$ and $-4$. Thus, the possible values of $b$ are $-16$ or $16$.
So the largest possible value of $b$ is $16$.

**Problem 2:**
Let the width of the rectangular garden be $w$, then the length would be $4w$.
The original area is:
$$A_{\text{original}} = l * w = 4w * w = 4w^2$$
If the width is increased by 5 feet, the new width is $w + 5$, and the new area is:
$$A_{\text{new}} = l * (w + 5) = 4w * (w + 5) = 4w^2 + 20w$$
Given that the total area of the garden is increased by 120 square feet:

$$A_{\text{new}} = A_{\text{original}} + 120$$
$$4w^2 + 20w = 4w^2 + 120$$

Solving for $w$:
$$20w = 120 \implies w = 6$$
So, the original area of the garden is:

$$A_{\text{original}} = 4w^2 = 4 * 36 = 144 \text{ square feet}$$

Problem 1 requires an understanding of quadratic equations, Vieta's formulas, and consideration of multiple possibilities, making it more challenging compared to Problem 2, which involves simpler algebraic manipulations with linear equations.

**Figure 10:** An example of using GPT-4 to evaluate the complexity of problems in `MathCodeInstruct`. "Problem 2" is in `MathCodeInstruct` and "Problem 1" is the problem from MATH that was used to generate "Problem 2".

---

**(a) Evaluation Prompt:**

Which problem is more difficult?

Problem 1: "Kara mixes cups of sugar and water in the ratio of 1:2 to make one liter of lemonade. Calculate the total number of cups of sugar used if she used 84 cups to make lemonade."

Problem 2: "A cylindrical water tank has two compartments. The first compartment is half the height of the second compartment. When filled, the first compartment holds 60 liters of water. If a pipe pumps water into the tank at a rate of 10 liters per minute and starts filling the second compartment, how many minutes will it take to fill the entire tank?"

You answer should be one of "Problem 1", "Problem 2" and "Tie".

---

**(b) Judgement from GPT-4:**

Let's solve each problem one by one and then determine which one is more difficult.

**Problem 1:**
Kara mixes cups of sugar and water in the ratio of $1 : 2$.
So, for every 1 cup of sugar, there are 2 cups of water.

The ratio of sugar to the total mixture is:

$$\text{Sugar : Total} = 1 : (1 + 2) = 1 : 3$$

She used 84 cups to make lemonade, so:

$$\text{Cups of sugar used} = \frac{1}{3} \times 84 = 28 \text{ cups}$$

**Problem 2:**
The first compartment is half the height of the second compartment but holds 60 liters of water.
So, the second compartment will hold $2 \times 60$ liters of water, which is 120 liters of water.

The total volume of the tank, including both compartments, is:

$$60 \text{ liters} + 120 \text{ liters} = 180 \text{ liters}$$

The pipe pumps water at a rate of 10 liters per minute, so to fill the entire tank, it will take:

$$\text{Time} = \frac{180 \text{ liters}}{10 \text{ liters/min}} = 18 \text{ minutes}$$

**Conclusion:**
Both problems can be solved with basic arithmetic and understanding of ratios, and neither requires advanced mathematical concepts. Therefore, it can be considered a "Tie" in terms of difficulty.

**Figure 11:** An example of using GPT-4 to evaluate the complexity of problems in `MathCodeInstruct`. "Problem 2" is in `MathCodeInstruct`, and "Problem 1" is the problem from GSM8K that was used to generate "Problem 2". **It is an example of a tie.**

(a) Evaluation Prompt:

Which problem is more difficult?

Problem 1: "When Jeffrey walks, for every three steps forward, he takes two steps backwards. Therefore, if the distance between the house and the mailbox is 66 steps, what is the total number of steps Jeffrey takes when he goes from the house to the mailbox?"

Problem 2: "For every four apples Jake eats, he plants three apple seeds. If at the end of the month he has consumed 24 apples, how many seeds has he planted?"

You answer should be one of "Problem 1", "Problem 2" and "Tie".

(b) Judgement from GPT-4:

Let's analyze each problem step by step:

**Problem 1:**
Every 3 steps forward and 2 steps back mean Jeffrey effectively moves 1 step for every 5 steps he takes (3 steps forward - 2 steps back = 1 step net movement).

To cover 66 steps:
Steps needed = (Total distance / Net movement per cycle) × Steps per cycle
Steps needed = (66 steps / 1 step) × 5 = 330 steps

**Problem 2:**
For every 4 apples Jake eats, he plants 3 seeds.
So, for 24 apples:
Seeds planted = (24 apples / 4) × 3 seeds = 6 × 3 = 18 seeds

To determine which is more difficult, it seems both problems involve similar basic math operations (division and multiplication).
However, the first problem introduces an extra layer of complexity with the two different types of movements (forward and backward) compared to the direct ratio in the second problem.

Given this analysis, **Problem 1** seems slightly more difficult than Problem 2 due to this added layer of understanding needed to solve it.

**Figure 12:** A example of using GPT-4 to evaluate the complexity of problems in `MathCodeInstruct` and it is an **uncommon** example that the problem from GSM8K is slightly more difficult than the interpolation problem. "Problem 2" is in `MathCodeInstruct` and "Problem 1" is the problem from GSM8K that was used to generate "Problem 2".

**(a) Prompt:**

Please create a new problem, following the given "Example" and the new problem should be harder than the "Example".

Example: "Josh's mom gives him $20 to go shopping at the mall. He buys a hat for $10 and a pencil for $2. Then he buys four cookies. If each cookie costs $1.25, how much money does Josh have left?"

**(b) According new problem generated by GPT-4:**

Rebecca's dad gives her $50 to spend at the amusement park. She buys a ticket for $25 and a souvenir for $6. After riding several rides, she decides to buy six balloons for her friends. If each balloon costs $2.50, how much money does Rebecca have left?

**(c) Evaluation Prompt:**

Which problem is more difficult?

Problem 1: "Josh's mom gives him $20 to go shopping at the mall. He buys a hat for $10 and a pencil for $2. Then he buys four cookies. If each cookie costs $1.25, how much money does Josh have left?"

Problem 2: "Rebecca's dad gives her $50 to spend at the amusement park. She buys a ticket for $25 and a souvenir for $6. After riding several rides, she decides to buy six balloons for her friends. If each balloon costs $2.50, how much money does Rebecca have left?"

You answer should be one of "Problem 1", "Problem 2" and "Tie".

**(d) Judgement from GPT-4:**

Let's solve both problems step-by-step:

**Problem 1:** Josh's initial amount: $20
He buys a hat: $20 - $10 = $10 left
He buys a pencil: $10 - $2 = $8 left
Cost of four cookies = 4 x $1.25 = $5
Amount after buying cookies: $8 - $5 = $3

So, Josh has $3 left.

**Problem 2:** Rebecca's initial amount: $50
She buys a ticket: $50 - $25 = $25 left
She buys a souvenir: $25 - $6 = $19 left
Cost of six balloons = 6 x $2.50 = $15
Amount after buying balloons: $19 - $15 = $4

So, Rebecca has $4 left.

In terms of difficulty, both problems involve subtraction and multiplication. However, Problem 2 has slightly larger numbers and one more multiplication operation than Problem 1. Hence, you could argue that:
**Answer:** Problem 2 is more difficult than Problem 1.

**Figure 13:** An example of using GPT-4 to create problems based only on the problems from GSM8K and then evaluate the complexity of the newly generated problems. "Problem 2" is the new problem, and "Problem 1" is the problem from GSM8K that was used to generate "Problem 2".

**(a) Prompt:**

Please create a new problem, following the given "Example" and the new problem should be harder than the "Example".

Example: "Kara mixes cups of sugar and water in the ratio of 1:2 to make one liter of lemonade. Calculate the total number of cups of sugar used if she used 84 cups to make lemonade."

**(b) According new problem generated by GPT-4:**

Liam mixes cups of honey, water, and lemon juice in the ratio of 2:5:3 to make one gallon of honey lemonade. Calculate the total number of cups of honey used if he used 120 cups to make honey lemonade.

**(c) Evaluation Prompt:**

Which problem is more difficult?

Problem 1: "Kara mixes cups of sugar and water in the ratio of 1:2 to make one liter of lemonade. Calculate the total number of cups of sugar used if she used 84 cups to make lemonade."

Problem 2: "Liam mixes cups of honey, water, and lemon juice in the ratio of 2:5:3 to make one gallon of honey lemonade. Calculate the total number of cups of honey used if he used 120 cups to make honey lemonade."

You answer should be one of "Problem 1", "Problem 2" and "Tie".

**(d) Judgement from GPT-4:**

Let's solve each problem to determine which one is more difficult:

**Problem 1:**
**If** the ratio of sugar to water is 1:2, then for every 3 cups (1 cup of sugar + 2 cups of water), 1 liter of lemonade is made.

**So**, if she used 84 cups to make lemonade, then:
Number of liters of lemonade made = 84 cups / 3 = 28 liters
**Therefore**, the total number of cups of sugar used is:
28 liters * 1 cup of sugar = 28 cups of sugar

**Problem 2:**
**If** the ratio of honey to water to lemon juice is 2:5:3, then for every 10 cups (2 cups of honey + 5 cups of water + 3 cups of lemon juice), 1 gallon of honey lemonade is made.

**So**, if he used 120 cups to make honey lemonade, then:
Number of gallons of honey lemonade made = 120 cups / 10 = 12 gallons

**Therefore**, the total number of cups of honey used is:
12 gallons * 2 cups of honey = 24 cups of honey

**Comparison:**
Both problems involve a similar approach to solve. Both require understanding ratios and proportionally dividing the given total cups by the sum of the ratio's components.

Verdict: Tie. Both problems are of similar difficulty.

**Figure 14:** An example of using GPT-4 to create problems based only on the problems from GSM8K and then evaluate the complexity of the newly generated problems. "Problem 2" is the new problem, and "Problem 1" is the problem from GSM8K that was used to generate "Problem 2".

**(a) Prompt:**

Please create a new problem, following the given "Example" and the new problem should be harder than the "Example".

Example: "Hillary always buys the Wednesday, Thursday and Friday editions of the local newspaper for $0.50 each. On Sunday, she spends $2.00 to get that copy. How much does she spend on the newspaper over 8 weeks?"

**(b) According new problem generated by GPT-4:**

Joseph purchases his favorite magazines on Monday, Tuesday, and Saturday every week. The Monday edition costs $1.25, the Tuesday edition costs $1.50, and the Saturday edition costs $2.75. Additionally, he buys a special weekly edition every Sunday for $3.50. How much does Joseph spend on purchasing his favorite magazines over 12 weeks?

**(c) Evaluation Prompt:**

Which problem is more difficult?

Problem 1: "Hillary always buys the Wednesday, Thursday and Friday editions of the local newspaper for $0.50 each. On Sunday, she spends $2.00 to get that copy. How much does she spend on the newspaper over 8 weeks?"

Problem 2: "Joseph purchases his favorite magazines on Monday, Tuesday, and Saturday every week. The Monday edition costs $1.25, the Tuesday edition costs $1.50, and the Saturday edition costs $2.75. Additionally, he buys a special weekly edition every Sunday for $3.50. How much does Joseph spend on purchasing his favorite magazines over 12 weeks?"

You answer should be one of "Problem 1", "Problem 2" and "Tie".

**(d) Judgement from GPT-4:**

Let's break down each problem:

**Problem 1:**
1. Hillary buys the newspaper on Wednesday, Thursday, and Friday for $0.50 each. So, the cost for these 3 days = 3 * $0.50 = $1.50
2. On Sunday, she spends $2.00.
3. The total she spends in a week = $1.50 + $2.00 = $3.50
4. Over 8 weeks, she spends = 8 * $3.50 = $28.00

**Problem 2:**
1. Joseph buys magazines on Monday for $1.25, Tuesday for $1.50, and Saturday for $2.75. So, the cost for these 3 days = $1.25 + $1.50 + $2.75 = $5.50
2. Additionally, he buys a special edition on Sunday for $3.50.
3. The total he spends in a week = $5.50 + $3.50 = $9.00
4. Over 12 weeks, he spends = 12 * $9.00 = $108.00

In terms of mathematical complexity, both problems involve the same operations: multiplication and addition. However, the second problem has more purchases and higher values involved, which could make it slightly more challenging for some people to solve. Also, the fact that Joseph's purchases extend over 12 weeks, as opposed to Hillary's 8 weeks, adds a layer of complexity.

So, based on the given information, the answer would be: **Problem 2**.

**Figure 15:** An example of using GPT-4 to create problems based only on the problems from GSM8K and then evaluate the complexity of the newly generated problems. "Problem 2" is the new problem, and "Problem 1" is the problem from GSM8K that was used to generate "Problem 2".

(a) Prompt:

Please create a new problem, following the given "Example" and the new problem should be harder than the "Example".

Example: "Peter has 15 birds. 1/3 are ducks. The rest are chickens and require special feed that costs $2 per bird. How much does it cost to feed the chickens?"

(b) According new problem generated by GPT-4:

Maria has 36 animals. 1/4 of them are rabbits. The rest are cats, and each cat needs a special food that costs $3 per cat per day. If Maria feeds the cats for 5 days, how much will she spend on cat food?

(c) Evaluation Prompt:

Which problem is more difficult?

Problem 1: "Peter has 15 birds. 1/3 are ducks. The rest are chickens and require special feed that costs $2 per bird. How much does it cost to feed the chickens?"

Problem 2: "Maria has 36 animals. 1/4 of them are rabbits. The rest are cats, and each cat needs a special food that costs $3 per cat per day. If Maria feeds the cats for 5 days, how much will she spend on cat food?"

You answer should be one of "Problem 1", "Problem 2" and "Tie".

(d) Judgement from GPT-4:

Let's break down the steps required to solve each problem:

**Problem 1:**
1. Calculate the number of ducks: $15 \times \frac{1}{3} = 5$.
2. Subtract the number of ducks from the total to find the number of chickens: $15 - 5 = 10$.
3. Calculate the cost to feed the chickens: $10 \times \$2 = \$20$.

**Problem 2:**
1. Calculate the number of rabbits: $36 \times \frac{1}{4} = 9$.
2. Subtract the number of rabbits from the total to find the number of cats: $36 - 9 = 27$.
3. Calculate the daily cost to feed the cats: $27 \times \$3 = \$81$.
4. Multiply the daily cost by 5 days: $5 \times \$81 = \$405$.

While both problems require a similar initial approach of finding the number of animals not explicitly stated, Problem 2 has an additional step of multiplying by the number of days to get the final cost.

Based on the number of steps and calculations required, **Problem 2** is slightly more difficult than Problem 1.

**Figure 16:** An example of using GPT-4 to create problems based only on the problems from GSM8K and then evaluate the complexity of the newly generated problems. "Problem 2" is the new problem, and "Problem 1" is the problem from GSM8K that was used to generate "Problem 2".

(a) Prompt:

Please create a new problem, following the given "Example" and the new problem should be harder than the "Example".

Example: "Zain has 10 more of each coin than Emerie. If Emerie has six quarters, seven dimes, and five nickels, how many coins does Zain have?"

(b) According new problem generated by GPT-4:

Xavier has 15 more of each coin than Elara. If Elara has nine quarters, eleven dimes, eight nickels, and three pennies, how many coins does Xavier have?

(c) Evaluation Prompt:

Which problem is more difficult?

Problem 1: "Zain has 10 more of each coin than Emerie. If Emerie has six quarters, seven dimes, and five nickels, how many coins does Zain have?"

Problem 2: "Xavier has 15 more of each coin than Elara. If Elara has nine quarters, eleven dimes, eight nickels, and three pennies, how many coins does Xavier have?"

You answer should be one of "Problem 1", "Problem 2" and "Tie".

(d) Judgement from GPT-4:

Let's break down each problem step by step:

**Problem 1:**
Emerie has: 6 quarters, 7 dimes, 5 nickels
For each type of coin, Zain has 10 more than Emerie.
Thus:
- Zain has 6 + 10 = 16 quarters
- Zain has 7 + 10 = 17 dimes
- Zain has 5 + 10 = 15 nickels
Total number of coins Zain has = 16 + 17 + 15 = 48 coins.

**Problem 2:**
Elara has: 9 quarters, 11 dimes, 8 nickels, 3 pennies
For each type of coin, Xavier has 15 more than Elara.
Thus:
- Xavier has 9 + 15 = 24 quarters
- Xavier has 11 + 15 = 26 dimes
- Xavier has 8 + 15 = 23 nickels
- Xavier has 3 + 15 = 18 pennies
Total number of coins Xavier has = 24 + 26 + 23 + 18 = 91 coins.

Now, let's evaluate the complexity:
Problem 1 has 3 types of coins and a difference of 10 for each coin.
Problem 2 has 4 types of coins and a difference of 15 for each coin.
Considering that "more difficult" might mean more steps or operations to solve, Problem 2 has more types of coins and a larger difference, so it requires slightly more computation.

Answer: **Problem 2** is more difficult.

**Figure 17:** An example of using GPT-4 to create problems based only on the problems from GSM8K and then evaluate the complexity of the newly generated problems. "Problem 2" is the new problem, and "Problem 1" is the problem from GSM8K that was used to generate "Problem 2".

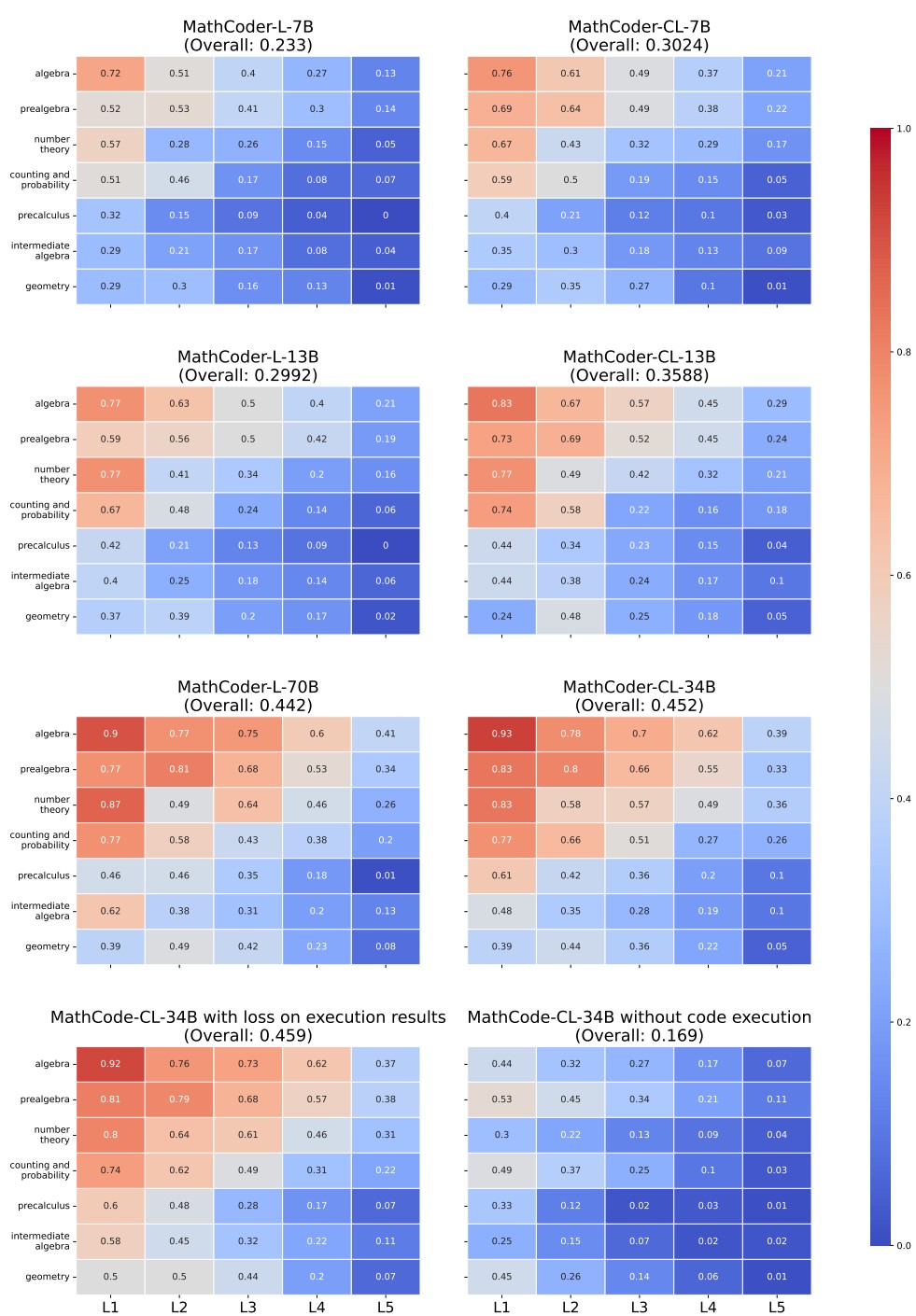

**Figure 18:** Performance comparison of MathCoeder-L (7B, 13B, 70B) and MathCoder-CL (7B, 13B, 34B) on the MATH dataset by levels and subjects. The improved accuracy from MathCoder-L to MathCoder-CL comes primarily from subjects that require precise calculations like algebra, number theory, and counting and probability.

**Problem: MATH/test/number-theory/1191**

<|user|><|text|> If $n$ is an integer, $1 \leq n \leq 2010$, how many fractions $\frac{n^2}{2010}$ yield repeating decimals? <|endofblock|><|endofmessage|>

**(a) CoT solution for MATH/test/number-theory/1191**

To determine whether a fraction $\frac{a}{b}$ produces a repeating decimal, you need to look at the denominator, $b$. A fraction will produce a repeating decimal if and only if, after reducing it to simplest form, the denominator has a prime factor other than 2 or 5.

Let's solve the problem step by step:

**Step 1:** Prime Factorization of 2010

$$2010 = 2 \times 3 \times 5 \times 67$$

**Step 2:** Repeating Decimal Criteria For $\frac{n^2}{2010}$ to be a repeating decimal, the denominator, after reducing the fraction to the simplest form, must have a prime factor other than 2 or 5.

**Step 3:** Counting Possible Values of $n$ To get a denominator with a prime factor other than 2 or 5 in reduced form, $n$ should not be divisible by 3, 67, or any multiple of them. So, we can find the possible values of $n$ as follows:

a) There are $2010/3 = 670$ numbers that are multiples of 3 between 1 and 2010.

b) There are $2010/67 = 30$ numbers that are multiples of 67 between 1 and 2010.

c) There are overlapping numbers which are multiples of $3 \times 67 = 201$. There are $2010/201 = 10$ such numbers.

**Step 4:** Applying the Inclusion-Exclusion Principle To avoid counting the overlapping numbers twice, we apply the inclusion-exclusion principle:

$$\text{Count} = 670 + 30 - 10 = 700 - 10 = 690$$

**Step 5:** Subtract from Total Possible Values of $n$ There are 2010 possible values for $n$ in total. To find the count of $n$ yielding repeating decimals:

$$2010 - 690 = 1320$$

So, there are 1320 ✗ fractions $\frac{n^2}{2010}$ that yield repeating decimals.

**(b) PoT solution for MATH/test/number-theory/1191**

```python
from sympy.ntheory import factorint

# Given range
n_range = range(1, 2011)
# Given denominator
denominator = 2010
# Initialize counter for fractions with
    repeating decimals
repeating_decimal_count = 0
# Factor the denominator to find the prime
    factorization
denominator_factors = factorint(denominator)
# Loop over the range to find the fractions
    with repeating decimals
for n in n_range:
    # Factor the numerator
    numerator_factors = factorint(n**2)
    # Check if each factor of the denominator
        is present in the numerator with at
        least the same power
    repeating_decimal = any(
        denominator_factors[p] >
        numerator_factors.get(p, 0) for p in
        denominator_factors)
    # If the fraction has repeating decimal,
        increase the counter
    if repeating_decimal:
        repeating_decimal_count += 1

repeating_decimal_count
```

>>> 2009 ✗

**(c) LCE solution for MATH/test/number-theory/1191**

<|assistant|> <|text|> Alright, for a fraction to yield a repeating decimal, its denominator (after simplification) should contain prime factors other than 2 and 5.

So, to solve this problem, we need to follow these steps:

1. Factorize 2010 to get its prime factors.
2. For each integer $n$ from 1 to 2010, compute $n^2$.
3. Divide $n^2$ by 2010 and simplify the fraction.
4. Check if the simplified denominator has prime factors other than 2 and 5.

If the denominator after simplification has only 2's and 5's as its prime factors, then it will yield a terminating decimal. Otherwise, it will yield a repeating decimal.

Let's proceed with these steps. <|endofblock|>

<|code|>

```python
from math import gcd

# Step 1: Factorize 2010 to get its prime
    factors.\\
def prime_factors(n):\\
    i = 2\\
    factors = {}\\
    while i * i <= n:\\
        if n % i:
            i += 1
        else:
            n //= i
            factors[i] = factors.get(i, 0) +
                1
    if n > 1:
        factors[n] = factors.get(n, 0) + 1
    return factors

factors_2010 = prime_factors(2010)

# Step 2, 3 and 4
count_repeating = 0
for n in range(1, 2011):
    n_square = n**2
    common_factor = gcd(n_square, 2010)
    denominator = 2010 // common_factor

    # Check if the simplified denominator has
        prime factors other than 2 and 5
    factors_denominator = prime_factors(
        denominator)
    if 2 in factors_denominator:
        del factors_denominator[2]
    if 5 in factors_denominator:
        del factors_denominator[5]

    if factors_denominator:  # if there are
        other prime factors left
        count_repeating += 1

count_repeating
```

<|endofblock|>

<|execution|> 2000 <|endofblock|>

<|text|> There are 2000 ✓ fractions of the form $\frac{n^2}{2010}$ that yield repeating decimals when $1 \leq n \leq 2010$.

<|endofblock|><|endofmessage|>

**Figure 19:** Example of CoT, PoT and LCE solution with special token. The problem is from the test set of MATH in number theory with id 1191. In contrast to CoT, which consists solely of natural language, and PoT, which consists solely of code, our LCE solution intertwines natural language, code, and execution results.

Problem: MATH/test/algebra/2477

<|user|><|text|> Consider the infinite arithmetic sequence $A$ with first term 5 and common difference $-2$. Now define the infinite sequence $B$ so that the $k^{th}$ term of $B$ is 2 raised to the $k^{th}$ term of $A$. Find the sum of all of the terms of $B$. <|endofblock|><|endofmessage|>

**(a) CoT solution for MATH/test/algebra/2477**

Alright. Let's start by understanding both sequences step by step.
**Sequence $A$**
This is an infinite arithmetic sequence. The first term, $a_1$, is 5, and the common difference, $d$, is -2.
So, we can describe the $k^{th}$ term of $A$ as:

$$a_k = 5 - 2(k - 1)$$
$$a_k = 5 - 2k + 2$$
$$a_k = 7 - 2k$$

**Sequence $B$**
The sequence $B$ is defined such that the $k^{th}$ term of $B$ is 2 raised to the $k^{th}$ term of $A$:

$$b_k = 2^{a_k} = 2^{7-2k}$$

**Sum of Sequence $B$**
Since $B$ is an infinite sequence, we are to find the sum of all terms of $B$:

$$S_B = 2^7 + 2^5 + 2^3 + 2^1 + \cdots$$

To find $S_B$, notice that it is a geometric series with the first term $2^7$ and common ratio $2^{-2} = \frac{1}{4}$.
The sum of an infinite geometric series can be found using the formula:

$$S = \frac{a}{1 - r}$$

where $a$ is the first term and $r$ is the common ratio.
Substitute the first term $2^7$ and the common ratio $\frac{1}{4}$ into the formula:

$$S_B = \frac{2^7}{1 - \frac{1}{4}} = \frac{2^7}{\frac{3}{4}} = \frac{2^7 \cdot \frac{4}{3}}{1} = \frac{2^9}{3}$$

Thus, the sum of all the terms of sequence $B$ is $\frac{2^9}{3}$. ✗

**(b) PoT solution for MATH/test/algebra/2477**

```
# Given first term and common difference of
    sequence A
first_term_A = 5
common_diff_A = -2

# Initialize sum of sequence B
sum_B = 0

# Loop over the terms of sequence A and
    calculate corresponding terms of
    sequence B,
# until the term of sequence A is non-
    positive.
kth_term_A = first_term_A
while kth_term_A > 0:
    kth_term_B = 2 ** kth_term_A
    sum_B += kth_term_B
    kth_term_A += common_diff_A

sum_B
```

$>>> 42$ ✗

**(c) LCE solution for MATH/test/algebra/2477**

<|assistant|><|text|> Sure. Let's solve the problem step by step.
**Step 1: Define Sequence $A$** Given that the first term of sequence $A$ is 5 and the common difference is -2, the $k^{th}$ term of $A$, denoted as $A_k$, can be found using the formula:

$$A_k = 5 - 2(k - 1)$$
$$A_k = 7 - 2k$$

**Step 2: Define Sequence $B$** The $k^{th}$ term of sequence $B$, denoted as $B_k$, is defined as 2 raised to the $k^{th}$ term of $A$:

$$B_k = 2^{A_k}$$

Substitute the value of $A_k$ into the equation for $B_k$:

$$B_k = 2^{7-2k}$$

**Step 3: Find Sum of Sequence $B$** To find the sum of all terms of sequence $B$, we need to find the sum to infinity of this sequence. Since the terms of sequence $B$ are getting smaller (because $2^{7-2k}$ decreases as $k$ increases), this is a geometric series.
The sum of an infinite geometric series can be found using the formula:

$$S = \frac{a}{1 - r}$$

Where:
- $S$ is the sum of the series,
- $a$ is the first term of the series,
- $r$ is the common ratio of the series.
Let's find the first term and common ratio of the sequence $B$ and then use them to find the sum of sequence $B$.
<|endofblock|>

<|code|>

```
# Given values
first_term_A = 5
common_difference_A = -2
base_B = 2

# Calculating the first term of sequence B
first_term_B = base_B ** first_term_A

# Calculating the common ratio of sequence B
# For k=1, A_k = 5, B_1 = 2^5
# For k=2, A_k = 3, B_2 = 2^3 => B_2 = B_1 /
    2^2
common_ratio_B = base_B ** (-2 * 1) # k=1, so
    2*1=2

# Calculating the sum of sequence B
sum_B = first_term_B / (1 - common_ratio_B)
first_term_B, common_ratio_B, sum_B
```

<|endofblock|>

<|execution|> (32, 0.25, 42.666666666666667) <|endofblock|>

<|text|>

**Step 3: Results** Thus, the sum of all of the terms of sequence $B$ is approximately 42.67. ✓ <|endofblock|><|endofmessage|>

**Figure 20:** Example of CoT, PoT and LCE solution with special token. The problem is from the test set of MATH in algebra with id 2477. In contrast to CoT, which consists solely of natural language, and PoT, which consists solely of code, our LCE solution intertwines natural language, code, and execution results.

**Problem: MATH/test/geometry/500**

<|user|><|text|> Points $A(0,0)$, $B(9,6)$ and $C(6,12)$ are vertices of triangle $ABC$. Point $D$ is on segment $AB$ such that $2(AD) = DB$, point $E$ is on segment $BC$ such that $2(BE) = EC$ and point $F$ is on segment $CA$ such that $2(CF) = FA$. What is the ratio of the area of triangle $DEF$ to the area of triangle $ABC$? Express your answer as a common fraction. <|endofblock|><|endofmessage|>

**CoT solution for MATH/test/geometry/500**

To find the coordinates of points $D$, $E$, and $F$, we can use the section formula. The section formula states that if a line divides two points $P(x_1, y_1)$ and $Q(x_2, y_2)$ in the ratio $m : n$, then the coordinates $(x, y)$ of the point dividing the line segment are given by:

$$x = \frac{mx_2 + nx_1}{m + n}$$

$$y = \frac{my_2 + ny_1}{m + n}$$

**Find coordinates of $D$**

$D$ divides $AB$ in the ratio $2 : 1$. Here, $A(0,0)$ and $B(9,6)$. So,

$$x_D = \frac{2 \cdot 9 + 1 \cdot 0}{2 + 1} = 6$$

$$y_D = \frac{2 \cdot 6 + 1 \cdot 0}{2 + 1} = 4$$

So, $D(6, 4)$.

**Find coordinates of $E$**

$E$ divides $BC$ in the ratio $2 : 1$. Here, $B(9,6)$ and $C(6,12)$. So,

$$x_E = \frac{2 \cdot 6 + 1 \cdot 9}{2 + 1} = 7$$

$$y_E = \frac{2 \cdot 12 + 1 \cdot 6}{2 + 1} = 10$$

So, $E(7, 10)$.

**Find coordinates of $F$**

$F$ divides $CA$ in the ratio $2 : 1$. Here, $C(6,12)$ and $A(0,0)$. So,

$$x_F = \frac{2 \cdot 0 + 1 \cdot 6}{2 + 1} = 2$$

$$y_F = \frac{2 \cdot 0 + 1 \cdot 12}{2 + 1} = 4$$

So, $F(2, 4)$.

Find the area ratio of $\triangle DEF$ to $\triangle ABC$ The area of a triangle formed by three vertices $P(x_1, y_1)$, $Q(x_2, y_2)$, and $R(x_3, y_3)$ is given by:

$$\text{Area} = \frac{1}{2} |x_1(y_2 - y_3) + x_2(y_3 - y_1) + x_3(y_1 - y_2)|$$

**1. Area of $\triangle ABC$:**

$$\text{Area}_{ABC} = \frac{1}{2} |0 \cdot (6 - 12) + 9 \cdot (12 - 0) + 6 \cdot (0 - 6)|$$

$$\text{Area}_{ABC} = \frac{1}{2} |0 + 108 + 36|$$

$$\text{Area}_{ABC} = 72 \text{ square units}$$

**2. Area of $\triangle DEF$:**

$$\text{Area}_{DEF} = \frac{1}{2} |6 \cdot (10 - 4) + 7 \cdot (4 - 4) + 2 \cdot (4 - 10)|$$

$$\text{Area}_{DEF} = \frac{1}{2} |36 + 0 - 12|$$

$$\text{Area}_{DEF} = 12 \text{ square units}$$

**Find the Ratio**

$$\text{Ratio} = \frac{\text{Area}_{DEF}}{\text{Area}_{ABC}}$$

$$\text{Ratio} = \frac{12}{72} = \frac{1}{6}$$

So, the ratio of the area of triangle $DEF$ to the area of triangle $ABC$ is $\frac{1}{6}$. ✗

**Figure 21:** Example of CoT solution. The problem is from the test set of MATH in geometry with id 500.

---

Problem: MATH/test/geometry/500

<|user|><|text|> Points $A(0,0)$, $B(9,6)$ and $C(6,12)$ are vertices of triangle $ABC$. Point $D$ is on segment $AB$ such that $2(AD) = DB$, point $E$ is on segment $BC$ such that $2(BE) = EC$ and point $F$ is on segment $CA$ such that $2(CF) = FA$. What is the ratio of the area of triangle $DEF$ to the area of triangle $ABC$? Express your answer as a common fraction. <|endofblock|><|endofmessage|>

---

PoT solution for MATH/test/geometry/500

```python
from sympy import symbols, solve, Point, Polygon, Rational

# Given points of the triangle ABC
A = Point(0, 0)
B = Point(9, 6)
C = Point(6, 12)

# Given that D is on segment AB such that 2(AD) = DB
# Therefore, AD:DB = 1:2
# Using section formula to find the coordinates of D
m1, m2 = 2, 1
D = Point((m1*B.x + m2*A.x)/(m1+m2), (m1*B.y + m2*A.y)/(m1+m2))

# Given that E is on segment BC such that 2(BE) = EC
# Therefore, BE:EC = 1:2
# Using section formula to find the coordinates of E
m1, m2 = 1, 2
E = Point((m1*C.x + m2*B.x)/(m1+m2), (m1*C.y + m2*B.y)/(m1+m2))

# Given that F is on segment CA such that 2(CF) = FA
# Therefore, CF:FA = 1:2
# Using section formula to find the coordinates of F
m1, m2 = 1, 2
F = Point((m1*A.x + m2*C.x)/(m1+m2), (m1*A.y + m2*C.y)/(m1+m2))

# Calculating the area of the triangles ABC and DEF
triangle_ABC = Polygon(A, B, C)
triangle_DEF = Polygon(D, E, F)

area_ABC = abs(triangle_ABC.area)
area_DEF = abs(triangle_DEF.area)

# Finding the ratio of the areas of triangle DEF to triangle ABC
ratio = Rational(area_DEF, area_ABC)
ratio
```

>>> 2/9 ✗

**Figure 22:** Example of PoT solution. The problem is from the test set of MATH in geometry with id 500.

Problem: MATH/test/geometry/500

<|user|><|text|> Points $A(0, 0)$, $B(9, 6)$ and $C(6, 12)$ are vertices of triangle $ABC$. Point $D$ is on segment $AB$ such that $2(AD) = DB$, point $E$ is on segment $BC$ such that $2(BE) = EC$ and point $F$ is on segment $CA$ such that $2(CF) = FA$. What is the ratio of the area of triangle $DEF$ to the area of triangle $ABC$? Express your answer as a common fraction. <|endofblock|><|endofmessage|>

LCE solution for MATH/test/geometry/500

<|assistant|><|text|> Let's solve the problem step by step:
**1. Find Points D, E, F:**
- Use the section formula to find the coordinates of the points $D$, $E$, and $F$.
- The section formula is:
$$(x, y) = \left( \frac{m \cdot x_2 + n \cdot x_1}{m + n}, \frac{m \cdot y_2 + n \cdot y_1}{m + n} \right)$$

- For point $D$, $m = 2$, $n = 1$, $x_1 = 0$, $y_1 = 0$, $x_2 = 9$, and $y_2 = 6$.
- For point $E$, $m = 2$, $n = 1$, $x_1 = 9$, $y_1 = 6$, $x_2 = 6$, and $y_2 = 12$.
- For point $F$, $m = 2$, $n = 1$, $x_1 = 6$, $y_1 = 12$, $x_2 = 0$, and $y_2 = 0$.
**2. Find the Area of the Triangles ABC and DEF:**
- The area of a triangle with vertices $(x_1, y_1)$, $(x_2, y_2)$, and $(x_3, y_3)$ is given by:
$$\text{Area} = \frac{|x_1(y_2 - y_3) + x_2(y_3 - y_1) + x_3(y_1 - y_2)|}{2}$$

**3. Find the Ratio of the Areas:**
- The ratio of the areas of triangles DEF and ABC is given by:
$$\text{Ratio} = \frac{\text{Area of DEF}}{\text{Area of ABC}}$$

Let's start by finding the coordinates of points D, E, and F using the section formula. <|endofblock|>

<|code|>

```
# Given points and ratios
A = (0, 0)
B = (9, 6)
C = (6, 12)
m1, n1 = 2, 1 # ratio for point D
m2, n2 = 2, 1 # ratio for point E
m3, n3 = 2, 1 # ratio for point F
# Using section formula to find coordinates of D, E, F
D = ((m1 * B[0] + n1 * A[0]) / (m1 + n1), (m1 * B[1] + n1 * A[1]) / (m1 + n1))
E = ((m2 * C[0] + n2 * B[0]) / (m2 + n2), (m2 * C[1] + n2 * B[1]) / (m2 + n2))
F = ((m3 * A[0] + n3 * C[0]) / (m3 + n3), (m3 * A[1] + n3 * C[1]) / (m3 + n3))

D, E, F
```

<|endofblock|>

<|execution|> ((6.0, 4.0), (7.0, 10.0), (2.0, 4.0)) <|endofblock|>

<|text|> The coordinates of the points are as follows:
- Point $D$ is located at $(6, 4)$
- Point $E$ is located at $(7, 10)$
- Point $F$ is located at $(2, 4)$
Now, let's proceed to find the areas of triangles $ABC$ and $DEF$ using the formula for the area of a triangle, and then find the ratio of these areas. <|endofblock|>

<|code|>

```
# Function to find the area of a triangle given its vertices
def triangle_area(v1, v2, v3):
    return 0.5 * abs(v1[0] * (v2[1] - v3[1]) + v2[0] * (v3[1] - v1[1]) + v3[0] * (v1[1] - v2[1]))
# Areas of triangles ABC and DEF
area_ABC = triangle_area(A, B, C)
area_DEF = triangle_area(D, E, F)
# Ratio of areas
ratio_areas = area_DEF / area_ABC

area_ABC, area_DEF, ratio_areas
```

<|endofblock|>

<|execution|> (36.0, 12.0, 0.3333333333333333) <|endofblock|>

<|text|> So, the ratio of the area of triangle $DEF$ to the area of triangle $ABC$ is $\frac{1}{3}$ ✔ or $0.3333$. <|endofblock|><|endofmessage|>

**Figure 23:** Example of LCE solution with special token. The problem is from the test set of MATH in geometry with id 500. In contrast to CoT, which consists solely of natural language, and PoT, which consists solely of code, our LCE solution intertwines natural language, code, and execution results.

