# OpenReview forum: "MathCoder: Seamless Code Integration in LLMs for Enhanced Mathematical Reasoning"
_ICLR.cc/2024/Conference — ICLR 2024 poster_

### Official Review · Reviewer_3VCG · 2023-10-28

**Soundness:** 3 good
**Presentation:** 3 good
**Contribution:** 2 fair
**Rating:** 6
**Confidence:** 4

**Summary:**

This paper proposes a new pipeline for training LLM for math problems with code integration. There are two main contributions in this paper. First, they use GPT-4 to do problem interpolation, which generates new problems between GSM8K (easy) and Math (hard) levels. Secondly, their dataset contains natural language for reasoning, code for execution and execution results, and they use these information to refine tune Llama-2 and code Llama. Their fine tuned model has achieved 45.2% on Math, which is much better than the previous results.

It is important to note that the labels for the interpolated problems were directly generated using their own fine tuned model based on CodeLlama with GPT-4 labels on GSM8K+Math, instead of using GPT-4.

**Strengths:**

I think the  main strength of this paper is it provides better results on many math datasets, including GSM8K and Math.

Originality: the main originality of this paper comes from the idea of doing problem interpolation, as well as using code integration. However, code integration is a common idea used in many math papers.

Quality: Good. This paper provides a clear pipeline of the algorithm, with detailed comparison with other methods.

Clarity: Good, it is easy to follow.

Significance: Mild, as stated below.

**Weaknesses:**

I think the main contribution of this paper is doing problem interpolation, but according to Table 4, the improvement of this idea is mild. In particular, in Table 2, the improvement of MathCoder-L and MathCoder-CL over existing methods on MATH is 22.4/22.5, but in Table 4, the help of problem interpolation is only 1.2. That means, almost all the improvement actually comes from the code integration part, which is the power from GPT-4 (with code interpreter).

I also find the use of Mathcoder-initial a bit confusing. Is the main purpose of this idea saving money? If so, it might be beneficial to explicitly point it out, otherwise it makes the whole training process unnecessarily complicated.

The fine tuning part for code integration is kind of straightforward, and there are many existing work using similar ideas. So I will not say it is an important contribution.

Overall speaking, I think the main contribution of this paper is "using GPT-4 to create a new dataset, augment the dataset using problem interpolation, and fine tune Llama using the created dataset". The augmentation is interesting, but with limited improvement. Therefore, I will say this paper has limited significance. I give weak accept mainly because I feel this is an important problem, and the authors provide a reasonably good solution.

**Questions:**

I do not have extra questions because the paper looks straightforward to understand. However, if I misunderstood anything about this paper, please do let me know and I will happy to update my score.

---

> ### Author Response · Authors · 2023-11-19
> **Response to Reviewer 3VCG (1/2)**
>
> **Q1**: I think the main contribution of this paper is problem interpolation, but according to Table 4, the improvement of this idea is mild. In particular, in Table 2, the improvement of MathCoder-L and MathCoder-CL over existing methods on MATH is 22.4/22.5, but in Table 4, the help of problem interpolation is only 1.2. That means almost all the improvement actually comes from the code integration part, which is the power from GPT-4 (with code interpreter).
>
> Overall, I think the main contribution of this paper is "using GPT-4 to create a new dataset, augment the dataset using problem interpolation, and fine-tune Llama using the created dataset". The augmentation is interesting but with limited improvement. Therefore, I will say this paper has limited significance. I give weak acceptance mainly because I feel this is an important problem, and the authors provide a reasonably good solution.
>
>
>
> **A1**: Thanks for your acknowledgment of the contribution of problem interpolation.
>
> Regarding the accuracy improvement, we would like to point out that in the ablation study in Table 4, the baseline is based on the model fine-tuned with CodeLlama-34B without including problem interpolation. This baseline has a significantly high performance. For example, it achieved 44.0% on the MATH dataset, comparable to GPT-4's performance. Although the baseline model has high performance, MathCoder-CL 34B with problem interpolation still improves the average accuracy by 4.0% across five datasets (Table 4, page 8). This demonstrates that the problem interpolation method effectively enhances the model's mathematical skills and generalization abilities.
>
> Additionally, we conducted further ablation studies to demonstrate the effectiveness of problem interpolation. These studies utilized the CodeLlama 7B model with varying numbers of interpolated problem samples. The results are as follows. As shown below, the average accuracy consistently improves with the increase in the number of problem interpolation data, demonstrating the effectiveness of our method. We have updated our paper to include the table below (Table 9, page 14).
>
>
> | Data Size              | GSM8K| MATH | SVAMP | Mathematics | SimulEq| Average |
> |------------------------|------|------|-------|-------------|--------|---------|
> | 49k+0                | 50.6  | 22.9  | 53.2  | 46.0        | 29.6   |**40.5**        |
> | 49k+11K              | 56.4  | 26.8  | 64.9  | 47.6        | 40.7   |**47.3** (+6.8) |
> | 49k+31K (baseline)   | 67.8  | 30.2  | 70.7  | 55.8        | 49.6   |**54.8** (+14.3)|
> | 49k+51K              | 68.0  | 32.6  | 70.9  | 60.1        | 52.7   |**56.9** (+16.4)|
>
> Moreover, we believe that problem interpolation can be explored further. As the baselines of WizardMath [1] and RFT [2] only use the GSM8k and MATH as seed data, we haven't explored problem interpolation on other types of datasets (e.g., SciBench [3]). We plan to explore it in future work.
>
> [1] Luo, Haipeng, et al. "Wizardmath: Empowering mathematical reasoning for large language models via reinforced evol-instruct." arXiv preprint arXiv:2308.09583 (2023).
>
> [2] Yuan, Zheng, et al. "Scaling relationship on learning mathematical reasoning with large language models." arXiv preprint arXiv:2308.01825 (2023).
>
>
> [3] Wang, Xiaoxuan et al. “SciBench: Evaluating College-Level Scientific Problem-Solving Abilities of Large Language Models.” ArXiv abs/2307.10635 (2023).

---

> ### Author Response · Authors · 2023-11-19
> **Response to Reviewer 3VCG (2/2)**
>
> **Q2**: I also find the use of Mathcoder-initial a bit confusing. Is the main purpose of this idea saving money? If so, it might be beneficial to explicitly point it out. Otherwise, it makes the whole training process unnecessarily complicated.
>
> **A2**: Sorry for the confusion about our choice of MathCoder-Initial over GPT-4 for constructing the training targets of D1 data.
>
> Firstly, a very important reason for using self-distillation is to save the cost. Using GPT-4 is expensive, and we have spent over $7,000 on building the training set without calculating the cost for model training and testing. We have mentioned in the original paper that it "reduces the cost compared to using GPT-4" (see page 5, section 4.1). We would be grateful if you could understand the choice.
>
> Secondly, this choice can demonstrate the efficacy of the interpolation method in solving the problem as we can enhance the math-solving accuracy using data generated by our own model, though the models are less powerful than GPT-4.
>
> Finally, using self-distillation further verifies that we have effectively trained our model to generate high-quality LCE solutions. This process further demonstrates our model's capability, showing that improvement through self-distillation is possible, which is also another major reason.
>
> To make the above reasons more sound, we have changed the original expression in page 4-5 of the revised paper. We hope this clears up your confusion on this issue.
>
>
> **Q3**: The fine-tuning part for code integration is kind of straightforward, and there are many existing works using similar ideas. So, I will not say it is an important contribution.
>
> **A3**: Thank you for your feedback. To the best of our knowledge, there is only one related work MAmmoTH [1], which also finetunes open-source models with math problems and solutions that contains natural language and code. However, their solutions consist of either only code or only natural language, while our LCE structure maintains the interleaved natural language and code simultaneously.
>
> Thanks to the advantage of the LCE block, we achieve better performance with fewer instruction data ( **80k** (ours) v.s. **260k** (MammoTH)). Therefore, the LCE-format solution is important for improving the mathematical reasoning of open-sourced models. The comparison is listed in the table below.
>
> |Method       | GSM8K | MATH | SVAMP | Mathematics| SimulEq | Average     |
> |:-----------|:-----:|:----:|:-----:|:----------:|:-------:|:-----------:|
> |MAmmoTH-34B  | 72.7  | 43.6 | 84.3  | 65.4       | 51.8    | 63.6        |
> |MathCoder-34B| 81.7  | 45.2 | 82.5  | 75.9       | 65.8    | 70.2 (+6.6)  |
> |MAmmoTH-70B  | 76.9  | 41.8 | 82.4  | 55.6       | 51.4    | 61.6        |
> |MathCoder-70B| 83.9  | 45.1 | 84.9  | 74.4       | 77.0    | 73.1 (+11.5)|
>
>
> [1] Yue, Xiang, et al. "Mammoth: Building math generalist models through hybrid instruction tuning." arXiv preprint arXiv:2309.05653 (2023).

---

> > ### Comment · Reviewer_3VCG · 2023-11-22
> >
> > Thank you for the explanation. I will not increase my score, but I will vote for accepting this paper.

---

> > > ### Author Response · Authors · 2023-11-22
> > > **Thanks for your response**
> > >
> > > Thank you for acknowledging our efforts.
> > >
> > > Best regards,
> > >
> > > Authors

---

### Official Review · Reviewer_Gqkc · 2023-10-31

**Soundness:** 3 good
**Presentation:** 2 fair
**Contribution:** 2 fair
**Rating:** 6
**Confidence:** 3

**Summary:**

The paper presents a method to fine-tune open-source models such as Llama-2 and CodeLlama for mathematical reasoning tasks. First, the authors propose a method to generate a large dataset of mathematical reasoning problems where each data sample consists of a natural language for reasoning, code for execution, and code execution results. They first obtain solutions for the seed datasets called the GSM8k and MATH using GPT-4 by interleaving the natural language, code, and code execution results for each question. Then, they fine-tune the CodeLlama-34B model using the seed data to produce their initial MathCoder model (MathCoder-Initial).

Second, to bridge the difficulty gap between grade-school-level problems from GSM8k and
challenging competition-level problems from MATH, they provide a novel prompting method to generate a dataset of intermediate-level problems by GPT-4. They give pairs of problems from GSM8k and MATH as prompts to GPT-4 and use the generated intermediate-level problems as additional training data with LCE solutions generated by the initial MathCoder model. The constructed dataset is called the MathCodeInstruct. Finally, they fine-tune the base Llama-2 and CodeLlama models using the MathCodeInstruct dataset to produce their final MathCoder models MathCoder-L and MathCoder-CL.

The authors evaluate the proposed models on five datasets including two in-domain datasets GSM8K and MATH, and three out-of-domain datasets SVAMP, Mathematics, and SimulEq. The results show that the proposed models outperform the open-source baselines on all datasets.

**Strengths:**

- The paper presents a novel method to fine-tune open-source models such as Llama-2 and CodeLlama for mathematical reasoning tasks. The proposed method is simple and effective.
- The paper presents a novel prompting method to generate a dataset of intermediate-level problems by GPT-4.

**Weaknesses:**

- I think it is really important to check whether or not the training dataset of GPT-4 contains datasets used in the paper. For instance, if GPT-4 is trained on one of the out-of-domain datasets considered in the paper, then it is very likely that GPT-4 copied the problems from the out-of-domain datasets and MathCoder used the copied problems for training. In this case, the results are not reliable because the baselines do not have access to the out-of-domain datasets.
- There is no theoretical analysis or guarantee of the proposed method.
- In Table 2, there is a typo as follows: Colsed-Source Model -> Closed-Source Model

**Questions:**

- I wonder if GPT-4 generated completely novel intermediate-level problems or just copied the problems from GSM8k and MATH (or from one of the out-of-domain datasets). If the latter is the case, I think it is not a fair comparison with the baselines because the baselines do not have access to the out-of-domain datasets.

---

> ### Author Response · Authors · 2023-11-19
> **Response to Reviewer Gqkc**
>
> **Q1**: I think it is really important to check whether or not the training dataset of GPT-4 contains datasets used in the paper. I wonder if GPT-4 generated completely novel intermediate-level problems or just copied the problems from GSM8k and MATH (or from one of the out-of-domain datasets). If the latter is the case, I think it is not a fair comparison with the baselines because the baselines do not have access to the out-of-domain datasets.
>
> **A1**: Thanks for your insightful comments. GPT-4 generated novel intermediate-level problems, and our training set and test set do not overlap; thus, the comparison with the baseline is fair.  Firstly, we evaluate the question through GPT-4 in Figure 4 in the paper, and 83.2% of interpolated problems are more challenging than GSM8K, and 95.6% of interpolated problems are simpler than MATH. Secondly, we directly check whether there is an intersection between the training set and the data of the five benchmarks. No two problems are exactly the same, so the new problems are not copied from the five datasets.
>
> To further address your question, we then use the Levenshtein distance [1,2] to detect whether there are similar questions.
> We sorted out the 5 most similar questions for each benchmark and found that even the most similar questions differed regarding decomposition ideas or calculation data [3]. Finally, it is worth mentioning that the baselines, WizardMath and RFT, also generate new questions using GPT-3.5 and GPT-4.
>
> [1] Levenshtein, Vladimir I. "Binary codes capable of correcting deletions, insertions, and reversals." Soviet physics doklady. Vol. 10. No. 8. 1966.
>
> [2] Chowdhery, Aakanksha, et al. "Palm: Scaling language modeling with pathways." arXiv preprint arXiv:2204.02311 (2022).
>
> [3] https://anonymous.4open.science/r/Reviewer-Gqkc_Q1--CA9E/dedup.json
>
>
> **Q2**: There is no theoretical analysis or guarantee of the proposed method.
>
> **A2**: Yes, the proposed MathCoder is an empirical study. Although it does not have theoretical support yet, we have empirically shown that the LCE block format can greatly improve the math reasoning accuracy on multiple public benchmarks (Table 2), and the proposed problem interpolation method can further improve math-solving models’ performance (Table 4, Table 8, and Table 9). Moreover, our method involves creating a diverse and high-quality math instruction dataset, followed by fine-tuning a large language model on this dataset. This approach is widely recognized as both common and effective [1,2].
>
> [1]. Li, X., Yu, P., Zhou, C., Schick, T., Zettlemoyer, L., Levy, O., ... & Lewis, M. (2023). Self-alignment with instruction backtranslation. arXiv preprint arXiv:2308.06259.
>
> [2]. Xu, C., Sun, Q., Zheng, K., Geng, X., Zhao, P., Feng, J., ... & Jiang, D. (2023). Wizardlm: Empowering large language models to follow complex instructions. arXiv preprint arXiv:2304.12244.
>
> **Q3**: In Table 2, there is a typo as follows: Colsed-Source Model -> Closed-Source Model
>
> **A3**: Thanks for your warm reminder. We have revised the typo.

---

> ### Author Response · Authors · 2023-11-22
> **Respectful Inquiry Before Discussion Deadline**
>
> Dear Reviewer Gqkc:
>
> Thanks again for your valuable time and comments. We have provided clarifications and explanations to address your concerns and followed the advice of all reviewers to improve our paper.
>
> With the discussion period drawing to a close,  please kindly let us know if anything is unclear. We truly appreciate this opportunity to improve our work and shall be most grateful for any feedback you could give us.
>
> Best regards,
>
> Authors

---

### Official Review · Reviewer_X1mb · 2023-11-03

**Soundness:** 2 fair
**Presentation:** 3 good
**Contribution:** 3 good
**Rating:** 5
**Confidence:** 4

**Summary:**

The paper proposes an approach called MathCoder to achieve better performance on math task. The proposed methods include steps:
- First,  use GPT4 to generate LCE solution (e.g. natural language (text) for reasoning L, code for execution C, and execution results E) for questions;
- Second, Finetune MathCoder-Initial 34B to learn to generate LCE solution
- Third, use GPT4 to generate data interpolation with a difficulty level between the easy and difficult datasets. Use MathCoder-Initial 34B to generate solution LCE solution
- Fourth, Finetune a smaller model (MathCoder-L or MathCoder-CL) on the augmented dataset

**Strengths:**

- First, the paper is quite well written, explained different steps clearly, discussion and ablation study are also interesting
- Second, the idea using GPT4 to interpolate questions to create augmentation is quite interesting
- Third, the paper designs the output format LCE, which includes reasoning, coding, execution, (with or without execution of the output by the code, say python python). this way encourages LLM to speak more, reasoning more, and potentially get good results
- Distillation of bigger model to smaller models give good performance.

**Weaknesses:**

- The proposed methods are comprises of multiple well known effective components (coding helps, execution, consistency, distallation). So the contribution of this paper may not be significant enough.
    - For example, using python code to solve math problem is well known to have much more advantages over text prompt (python code is more precise than text). as indicated by paper "Program-aided Language Models" and many other paper
    - Another example,  including execution of the  python code can improve performance, as it avoids LLM hallucinate on the equations, such as tool use or plugin paper.
    - Distillation of bigger models on GPT-4's solution isn't quite new.

- One of the main contribution of the paper is LCE solution format. However, even we let GPT4 generate solution with demonstration of solution with both reasoning, math, code, it may not generate well-moduled solution with clear blocked  <|text|> <|code|> and <|execution|>. it may more likely to have intersected solution mixing the three together. how to maintain the well structured LCE solution? If the method doesn't relay on the well moduled solution LCE, then the method simply boils down to distillation on GPT4's solution? what will the finetuning results be if the solution format is only python coding, which probably be the major performance contributior?

**Questions:**

1 typo: From the comparison between Tab. 5 (#1) and Tab. 5 (#2), we can see that Tab. 5 (#1) outperforms Tab. 5 (#2) across all five datasets, showing an improvement of 34.0% in the average accuracy score.

    - #2 is better?

2 MathCoder-Initial already has an accuracy of 77.3% on GSM8K and 44.0% on MATH. MathCoder is the model used to generate ground truth solution for supervised finetuning smaller models. How do we explain why the performance on the distilled smaller models have better performance in Table 2?

3 During supervised finetuning, can you say more on how do you implement "cross-question masking"? how does it help of the performance compared with not using it?
" In order to make the training more efficient, several
instances are concatenated together to form a single input, while cross-question masking is used to
ensure only tokens in the same instance are visible."

4. Another interesting experiment to show the improvement of the interpolation augmentation is the changes of performance with the size of the augmentation. Are they always help? when will it saturate?

---

> ### Author Response · Authors · 2023-11-19
> **Response to Reviewer X1mb (1/3)**
>
> **Q1**: The proposed methods are comprised of multiple well-known effective components (coding helps, execution, consistency, distillation). So, the contribution of this paper may not be significant enough. (three examples)
>
> **A1**: Thanks for your comments; we justify the difference with the examples you mentioned and highlight our contributions as follows:
>
> **Relationship with the generation and execution of the Python code to solve math problems:**
> Yes, it is well known that existing work utilizing code execution to improve reasoning ability, such as PoT [1], PAL [2],  etc. However, those works are built on the closed-sourced models (e.g., GPT-3.5, GPT-4), while this paper focuses on improving the mathematical reasoning of open-source models (Llama2 and CodeLlama). Furthermore, typical code execution work mostly uses only one block of code (PoT [1], PAL [2]). In contrast, our method constructs the LCE format solutions, which interleaves the natural language and code chain for multiple times. This format can improve the mathematical reasoning of open-source models. We add the ablation study based on CodeLlama-7B, as shown in the table below. The LCE format solution results in a 17.9\% accuracy gain compared to code-only answers with program execution. In summary, no existing work presents the LCE-style instruction dataset for solving math problems. We add the ablation study in the revised version (Table 5, page 9) and present the results as follows:
>
>
> | Data Format          | GSM8K | MATH | SVAMP | Mathematics | SimulEq | Average |
> |:--------------------|:-----:|:----:|:-----:|:-----------:|:-------:|:-------:|
> | LCE (ours)       | 67.8  | 30.2 | 70.7  |   55.8     | 49.6   | 54.8    |
> | Only Code            | 50.2  | 20.2 | 61.6  | 39.8       | 12.8    | 36.9 **(-17.9)** |
>
>
>
> [1]. Chen W, Ma X, Wang X, et al. Program of thought prompting: Disentangling computation from reasoning for numerical reasoning tasks[J]. Transactions on Machine Learning Research. (https://openreview.net/forum?id=YfZ4ZPt8zd)
>
> [2]. Gao, L., Madaan, A., Zhou, S., Alon, U., Liu, P., Yang, Y., ... & Neubig, G. (2023, July). Pal: Program-aided language models. In International Conference on Machine Learning (pp. 10764-10799). PMLR.
>
>
> **Relationship with distillation of bigger models on GPT-4's solution:**
>
> Distillation from bigger models (fine-tuned from GPT-4's solution) is not our main contribution; we use the distillation method to validate that the MathCoder models with LCE block can self-generate solution data without assistance from GPT-4. This is because GPT-4 is expensive. The MathCoder can maintain the `<|text|>`, `<|code|>` and `<|execution|>` format and the solution from MathCoder can further improve mathematical reasoning capability of itself.
>
>
> **Q2**: One of the main contributions of the paper is the LCE solution format. However, even if we let GPT4 generate a solution with a demonstration of a solution with both reasoning, math, and code, it may not generate a well-moduled solution with clear blocked `<|text|>`, `<|code|>` and `<|execution|>`. It may be more likely to have an intersected solution mixing the three together. How do you maintain a well-structured LCE solution?
>
> **A2**: We can guarantee that all samples in our train set have well-moduled solutions with clear blocked `<|text|>`, `<|code|>` and `<|execution|>`. To maintain well-structured LCE solutions, we constructed the original datasets with the following steps:
>
> Firstly, we prompted GPT-4 to solve the given math problem using code step by step and put the final answer with a specific prefix.
>
> Secondly, we extracted the code in the response and executed it in our local environment. The texts after the extracted code were discarded to maintain the LCE format.
>
> Thirdly, we concatenated the execution results to the prompt and model output and let GPT-4 continue to generate the problem-solving process.
>
> We repeated the three processes until GPT-4 no longer outputs code and we detected the final answer. In this way, we can get a well-structured LCE solution.
>
>
> **Q3**: If the method doesn't relay on the well moduled solution LCE, then the method simply boils down to distillation on GPT4's solution?
>
> **A3**: Our model’s final performance heavily relies on the proposed LCE format solutions. We introduced special tokens `<|text|>`, `<|code|>` and `<|execution|>` during training, and consequently, we can control the model to generate strict LCE format output during decoding.

---

> ### Author Response · Authors · 2023-11-19
> **Response to Reviewer X1mb (2/3)**
>
> **Q4**: What will the fine-tuning results be if the solution format is only Python coding, which will probably be the major performance contributor?
>
> **A4**: Thank you for your interest in our work. The average accuracy drops by 17.9 points when we use code-only format solutions to train the model. The results are presented in the table below. The experiments in this ablation study are based on CodeLlama 7B. This shows that the LCE format is an important performance contributor. We have updated the paper to include the results in Table 5 (page 9).
>
> | Data Format          | GSM8K | MATH | SVAMP | Mathematics | SimulEq | Average |
> |:--------------------|:-----:|:----:|:-----:|:-----------:|:-------:|:-------:|
> | LCE (ours)       | 67.8  | 30.2 | 70.7  |   55.8     | 49.6   | 54.8    |
> | Only Code            | 50.2  | 20.2 | 61.6  | 39.8       | 12.8    | 36.9 **(-17.9)** |
>
>
> **Q5**: typo: From the comparison between Tab. 5 (#1) and Tab. 5 (#2), we can see that Tab. 5 (#1) outperforms Tab. 5 (#2) across all five datasets, showing an improvement of 34.0% in the average accuracy score.
>
> **A5**: Thanks for your warm reminder; we have revised the sentence.
>
> **Q6**: MathCoder-Initial already has an accuracy of 77.3% on GSM8K and 44.0% on MATH. MathCoder is the model used to generate ground truth solutions for the supervised fine-tuning of smaller models. How do we explain why the performance of the distilled smaller models has better performance in Table 2?
>
> **A6**: Thank you for your comments. The performance of the distilled smaller models is presented in the table below. As can be seen, their accuracy on GSM8K and MATH is all lower than MathCoder-Initial. We highlight that MathCoder-Initial is based on **CodeLlama-34B**.
>
> | Model            | GSM8K | MATH |
> | ---------------- | :-----: | :----: |
> |**MathCoder-Initial** | **77.3** | **44.0** |
> | MathCoder-L-7B   | 64.2  | 23.3 |
> | MathCoder-L-13B  | 72.6  | 29.9 |
> | MathCoder-CL-7B  | 67.8  | 30.2 |
> | MathCoder-CL-13B | 74.1  | 35.9 |

---

> ### Author Response · Authors · 2023-11-19
> **Response to Reviewer X1mb (3/3)**
>
> **Q7**: During supervised fine-tuning, can you say more about how you implement "cross-question masking"? How does it help with performance compared to not using it?
>
> **A7**: The average token length of our 80k train set is 653 and 98.5% of the sample lengths are shorter than 2048. So, we set the token length as 2048. If we don’t concatenate different question-answer pairs together, each training sample would only contain one (question, solution) pair, and more than two-thirds of the tokens would be wasted on padding tokens. To address this issue, we concatenate several (question, solution) pairs into one sample to maximize the utilization of GPU[1][2]. Under this setting, the training can be sped up by 2.5x.
>
> If we didn't modify the masking, the model could see prior questions and corresponding solutions when predicting the target solutions, similar to few-shot prompting, which would differ from the inference stage. So, tokens from previous samples are masked to eliminate distractions when predicting one question's solution. The masking is in the shape of:
> ```
> 1 0 0 0 0 0 0 0 0
> 1 1 0 0 0 0 0 0 0
> 0 0 1 0 0 0 0 0 0
> 0 0 1 1 0 0 0 0 0
> 0 0 1 1 1 0 0 0 0
> 0 0 1 1 1 1 0 0 0
> 0 0 0 0 0 0 1 0 0
> 0 0 0 0 0 0 1 1 0
> 0 0 0 0 0 0 1 1 1
> ```
> In addition, the Llama models use Rotary Position Embedding, which means the attention scores are only related to the relative distance between two tokens. So, there is no gap between training and inference when concatenating several samples and using cross-question masking.
>
> The following table shows the influence of cross-question masking on performance. We adopted this technique mainly to improve training speed and efficiency. Cross-question masking is relatively straightforward, and we do not claim it as part of our contribution (see page 2 and page 3, Introduction).
>
> |  Samples Data              | GSM8K | MATH  | SVAMP | Mathematics | SimulEq| Average        |
> |:--------------------------|:-----:|:-----:|:-----:|:-----------:|:------:|:--------------:|
> | w/ cross-question masking  | 67.8  | 30.2  | 70.7  | 55.8        | 49.6   |**54.8**        |
> | w/o cross-question masking | 63.5  | 31.6  | 70.4  | 50.1        | 51.95  |**52.5** (-2.3) |
>
> [1] Liu B, Chen C, Liao C, Gong Z, Wang H, Lei Z, Liang M, Chen D, Shen M, Zhou H, Yu H. MFTCoder: Boosting Code LLMs with Multitask Fine-Tuning. arXiv preprint arXiv:2311.02303. 2023 Nov 4.
>
> [2] https://github.com/bigcode-project/starcoder/blob/main/chat/train.py#L190
>
> **Q8**: Another interesting experiment showing the improvement of the interpolation augmentation is the change in performance with the augmentation size. Do they always help? When will they saturate?
>
> **A8**: We thank the reviewer for raising this important point. Following your suggestion, we added experiments with different augmented sample sizes. The results are presented in the table below. As can be seen, there is a consistent improvement in average accuracy as the number of samples generated using problem interpolation increases. We have updated the paper to include this table (Table 9).
>
> |  Samples Data        | GSM8K | MATH  | SVAMP | Mathematics | SimulEq| Average        |
> |:--------------------|:-----:|:-----:|:-----:|:-----------:|:------:|:--------------:|
> | 49k+0                | 50.6  | 22.9  | 53.2  | 46.0        | 29.6   |**40.5**        |
> | 49k+11K              | 56.4  | 26.8  | 64.9  | 47.6        | 40.7   |**47.3** (+6.8) |
> | 49k+31K (baseline)   | 67.8  | 30.2  | 70.7  | 55.8        | 49.6   |**54.8** (+14.3)|
> | 49k+51K              | 68.0  | 32.6  | 70.9  | 60.1        | 52.7   |**56.9** (+16.4)|
>
> Additionally, we believe there is space for further exploration. For example, we could use three or more different types of seed questions, further increasing the diversity of the generated questions. We leave this to future work.

---

> ### Author Response · Authors · 2023-11-22
> **Respectful Inquiry Before Discussion Deadline**
>
> Dear Reviewer X1mb,
>
> Thanks again for your great efforts in reviewing this paper! With the discussion period drawing to a close, we expect your feedback and thoughts on our reply. We put a significant effort into our response, with several new adjustments and discussions. We sincerely hope you can consider our reply in your assessment.
>
> We look forward to hearing from you, and we can further address unclear explanations and remaining concerns, if any.
>
> Best,
>
> Authors

---

### Official Review · Reviewer_VXbY · 2023-11-05

**Soundness:** 3 good
**Presentation:** 3 good
**Contribution:** 3 good
**Rating:** 6
**Confidence:** 5

**Summary:**

This work presents MathCoder, a family of open-source LLMs for math problem solving. For training data construction, they first use GPT-4 to generate solutions to GSM8K and MATH problems, where the solutions contain natural language, code and execution results. This dataset is used to train MathCoder-Initial, which is finetuned from Llama-2 or CodeLlama. Afterward, the authors design problem interpolation prompting to generate new training problems with the difficulty between GSM8K and MATH, and use MathCoder-Initial to generate solutions for further finetuning. They evaluate MathCoder on several math benchmarks, and show that MathCoder outperforms other open-source LLMs.

**Strengths:**

1. Improving opensource LLMs is a good research topic. MathCoder noticeably outperforms other opensource LLMs, especially on challenging math benchmarks such as MATH.

2. The ablation studies show some interesting observations, e.g., comparison between models based on Llama-2 and CodeLlama, and predicting the execution results during training degrades the performance.

3. Besides the MathCoder checkpoints, MathCodeInstruct can also be valuable data for the community.

**Weaknesses:**

This work presents a complete training framework that achieves impressive performance. However, some design choices lack related ablation studies and explanation.

1. To construct the training targets for D1 data, why using MathCoder-Initial instead of GPT-4? How is the performance if GPT-4 is used for generating the solutions? My hypothesis is that training on MathCoder-Initial's own predictions can amplify the model's own prediction mistakes, and it is helpful to explain more about why this can still improve the performance.

2. The authors emphasize the importance of having natural language, code and execution results altogether in each problem solution. It would be good to have an ablation where the solutions only contain natural language or code, then we can check how much improvement is obtained by adding different components together in one solution.

3. To understand the importance of problem interpolation prompting, it would be helpful to have an ablation where the generated data is a mixture of data generated with GSM8K or MATH as the seed data, instead of using interporation.

4. It is helpful to have an analysis on performance with different number of generated training samples, so that we can better understand how much improvement is possible with this approach.

**Questions:**

1. Please explain why using MathCoder-Initial instead of GPT-4 for generating the training targets, and what is the performance if GPT-4 is used for generating the solutions.

2. It would be good to have an ablation where the solutions only contain natural language or code, then we can check how much improvement is obtained by adding different components together in one solution.

3. It would be helpful to have an ablation where the generated data is a mixture of data generated with GSM8K or MATH as the seed data, instead of using interporation.

4. It is helpful to have an analysis on performance with different number of generated training samples, so that we can better understand how much improvement is possible with this approach.

---

> ### Author Response · Authors · 2023-11-19
> **Response to Reviewer VXbY (1/2)**
>
> We sincerely thank you for your comments and appreciation of our work. Our responses to your concerns are demonstrated below:
>
> **Q1**: To construct the training targets for D1 data, why using MathCoder-Initial instead of GPT-4?
>
> **A1**: Sorry for the confusion about our choice of MathCoder-Initial over GPT-4 for constructing the training targets of D1 data.
>
> Firstly, a very important reason for using self-distillation is to save the cost. Using GPT-4 is expensive, and we have spent over $7,000 on building the training set without calculating the cost for model training and testing. We have mentioned in the original paper that it "reduces the cost compared to using GPT-4" (see page 5, section 4.1). We would be grateful if you could understand the choice.
>
> Secondly, this choice can demonstrate the efficacy of the interpolation method in solving the problem as we can enhance the math-solving accuracy using data generated by our own model, though the models are less powerful than GPT-4.
>
> Finally, using self-distillation further verifies that we have effectively trained our model to generate high-quality LCE solutions. This process further demonstrates our model's capability, showing that improvement through self-distillation is possible, which is also another major reason.
>
> To make the above reasons more sound, we have changed the original expression in page 4-5 of the revised paper. We hope this clears up your confusion on this issue.
>
> **Q2**: How is the performance if GPT-4 is used for generating the solutions?
>
> **A2**: Following your suggestion, we subsequently used GPT-4 to generate the 31k data with some extra funding, creating a comparative dataset. We trained a version of CodeLlama-7/34B with this dataset, facilitating a direct comparison with our original model. The results are detailed in the table below. We have updated our paper to include the results in Table 10 (page 15).
>
> | Base Model      | Data  | GSM8K| MATH | SVAMP | Mathematics | SimulEq | Average          |
> |:---------------|:-----------------------|:----:|:----:|:-----:|:-----------:|:-------:|:----------------:|
> | CodeLlama-7B    | 49k (GPT-4)+31k (MathCoder-Initial)         | 67.8 | 30.2 | 70.7  | 55.8        | 49.6    | 54.8             |
> | CodeLlama-7B    | **80k (GPT-4)**                   | 68.4 | 31.2 | 76.3  | 61.6        | 52.5    | 58.0 **(+3.2)**  |
> | CodeLlama-34B   | 49k (GPT-4)+31k (MathCoder-Initial)         | 81.7 | 45.2 | 82.5  | 75.9        | 65.8    | 70.2             |
> | CodeLlama-34B   | **80k (GPT-4)**                   | 82.2 | 47.6 | 84.1  | 79.2        | 69.7    | 72.6 **(+2.4)**  |
>
> As expected, using GPT-4 generated solutions leads to increased performance. With the 31k data generated from GPT-4 instead of MathCoder-Initial, our MathCoder- CL-7B and CL-34B's performances on all benchmarks can further increase. We have updated our paper with the results and the corresponding analysis in Appendix E.
>
> **Q3**: My hypothesis is that training on MathCoder-Initial's own predictions can amplify the model's own prediction mistakes, and it is helpful to explain more about why this can still improve the performance.
>
> **A3**: The experiments don't show the issue of error amplification by our MathCoder-Initial. One reason is that the MathCoder-Initial model is based on CodeLlama-34B, and it has already achieved an accuracy of 77.3% on GSM8K and 44.0% on MATH, demonstrating its inherent problem-generation ability. More importantly, to avoid amplification of the model's own prediction mistakes, we didn't take all the data generated by MathCoder-Initial. Instead, "we generate n different LCE solutions as depicted in (Wang et al., 2023a) for each new problem with our initial MathCoder models, keeping only those solutions for which all n answers match (n is set to 3 in this paper), thus ensuring our dataset's quality." We have revised the related paragraph to make it clearer.

---

> ### Author Response · Authors · 2023-11-19
> **Response to Reviewer VXbY (2/2)**
>
> **Q4**: It would be good to have an ablation where the solutions only contain natural language or code; then, we can check how much improvement is obtained by adding different components together in one solution.
>
> **A4**: We thank your suggestion for this meaningful ablation study. We list the results in the table below. Our new ablation study shows that the LCE format solutions outperform code-only format solutions by 17.9 points, and natural-language-only format solutions by 30.0 points.
>
> WizardMath 7B [1] serves as the performance of natural-language-only format solutions, as we do not have enough natural-language-only format data to train a new model. Results of WizardMath 7B are taken from [1]. We have updated our paper to include the results in Table 5 and Table 6 (page 9).
>
> LCE format v.s. Code-only (Base model: CodeLlama 7B)
>
> | Data Format          | GSM8K | MATH | SVAMP | Mathematics | SimulEq | Average |
> |:--------------------|:-----:|:----:|:-----:|:-----------:|:-------:|:-------:|
> | LCE (ours, 80k)       | 67.8  | 30.2 | 70.7  |   55.8     | 49.6   | 54.8    |
> | Only Code (80k)           | 50.2  | 20.2 | 61.6  | 39.8       | 12.8    | 36.9 **(-17.9)** |
>
>
> LCE format v.s. Natural-language-only (Base model: Llama 7B)
> | Data Format          | GSM8K | MATH | SVAMP | Mathematics | SimulEq | Average |
> |:--------------------|:-----:|:----:|:-----:|:-----------:|:-------:|:-------:|
> | LCE (ours, 80k)       | 64.2  | 23.3 | 71.5  |  46.9     | 47.5   | 50.7    |
> | WizardMath 7B [1] (96k, **NL**-format) | 54.9  | 10.7 | 36.1  | 9.3        | 12.8    | 24.8 **(-25.9)**|
>
>
>
> [1] Luo, Haipeng, et al. "Wizardmath: Empowering mathematical reasoning for large language models via reinforced evol-instruct." arXiv preprint arXiv:2308.09583 (2023).
>
> **Q5**: To understand the importance of problem interpolation prompting, it would be helpful to have an ablation where the generated data is a mixture of data generated with GSM8K or MATH as the seed data instead of using interpolation.
>
> **A5**: Thank you for suggesting this interesting ablation study. To better understand the importance of the proposed problem interpolation via prompting, we have generated 31K data samples using GSM8K or MATH as the seed data with equal portions, without employing problem interpolation. Using this data, combined with the original D0 dataset, we trained the CodeLlama-7B and compared it with the result using problem interpolation. We present the performances in the table below.
>
> | seed_data                           | GSM8K | MATH | SVAMP | Mathematics | SimulEq | Average |
> |-----------------------------------|:-----:|:----:|:-----:|:-----------:|:-------:|:-------:|
> | w/  interporation (MathCoder-CL-7B) | 67.8  | 30.2 | 70.7  | 55.8        | 49.6    | 54.8    |
> | w/o interporation                   | 61.9  | 29.1 | 70.9  | 50.5        | 43.4    | 51.2 **(-3.6)** |
>
>
>
> As shown above, the results with problem interpolation surpass those obtained without it in 4 out of 5 datasets, leading to a 3.6% accuracy improvement in average across the five datasets. This indicates that problem interpolation enhances accuracy of in-domain datasets such as GSM8K and MATH and aids out-of-domain datasets like Mathematics and SimulEq. Such generalization likely stems from the method's ability to bridge the gap between the more challenging MATH problems and the relatively simpler GSM8K problems.
>
> We have updated our paper to include the results in Table 8 (page 14).
>
>
> **Q6**: It is helpful to have an analysis of performance with different numbers of generated training samples so that we can better understand how much improvement is possible with this approach.
>
> **A6**: Thank you for suggesting the analysis. We trained the CodeLlama 7B model with varying numbers of interpolated problem samples to answer the question. As shown by the results below, the average accuracy consistently improves with the increase in the number of problem interpolation data, which demonstrates the effectiveness of our method. We have updated our paper to include the results in Table 9 (page 14).
>
> | Data Size              | GSM8K| MATH | SVAMP | Mathematics | SimulEq| Average |
> |:------------------------|:------:|:------:|:-------:|:-------------:|:--------:|:---------:|
> | 49k+0                | 50.6  | 22.9  | 53.2  | 46.0        | 29.6   |**40.5**        |
> | 49k+11K              | 56.4  | 26.8  | 64.9  | 47.6        | 40.7   |**47.3** (+6.8) |
> | 49k+31K   | 67.8  | 30.2  | 70.7  | 55.8        | 49.6   |**54.8** (+14.3)|
> | 49k+51K              | 68.0  | 32.6  | 70.9  | 60.1        | 52.7   |**56.9** (+16.4)|

---

> ### Author Response · Authors · 2023-11-22
> **Respectful Inquiry Before Discussion Deadline**
>
> Dear Reviewer VXbY,
>
> As the deadline for the author-reviewer discussion phase is nearing, we respectfully ask if our rebuttal has correctly addressed the concerns raised in your review.
>
> Your insightful feedback regarding the ablation studies on generated data and solutions has been invaluable in refining our work. We have endeavored to address each of these points meticulously in our rebuttal. We are eager to know if our responses and subsequent revisions have met your expectations and solved the issues you pointed out.
> We sincerely recognize the demands of your time and would appreciate any further feedback on our rebuttal. Your expertise is vital for the review process and enriches our ongoing learning and growth in this field.
>
> Thank you once again for your time and constructive insights. We eagerly look forward to your response at your earliest convenience.
>
> Best regards,
>
> Authors

---

> > ### Comment · Reviewer_VXbY · 2023-11-23
> >
> > I thank the authors for the response and paper vision with more ablation studies. I keep my initial review score.

---

> > > ### Author Response · Authors · 2023-11-23
> > > **Thanks for your response**
> > >
> > > Thank you for your positive comments.
> > >
> > > Best regards,
> > >
> > > Authors

---

### Author Response · Authors · 2023-11-20
**Paper Update Summary**

Dear all reviewers and chairs,

Thanks to all the chairs and reviewers for constructive suggestions and comments; we summarized the details of the revised paper.


1. Polish the reason for using self-distillation for better clarity in section 2.1. **(Reviewer VXbY, Reviewer 3VCG)**
2. Clarification of data contamination in section 2.1. **(Reviewer Gqkc)**
3. Ablation study of generating data samples using GSM8K or MATH as the seed data separately in section 3.3 and in Table 8. **(Reviewer VXbY)**
4. Ablation study of LCE solutions compared to code-only or natural-language-only solutions in section 3.3 and in Table 5 and Table 6. **(Reviewer VXbY, Reviewer X1mb)**
5. Ablation study of self-distillation in Table 11. **(Reviewer VXbY)**
6. Ablation study of different numbers of problem interpolation samples in Table 9. **(Reviewer VXbY, Reviewer X1mb, Reviewer 3VCG)**

Apart from the issues addressed above, we revised our paper about consistency and readability.

---

### Meta-Review · Area_Chair_vALw · 2023-12-11

**Metareview:**

This work presents MathCoder, a family of open-source LLMs based on Llama and Code Llama and MathCodeInstruct, GPT-4 generated solutions to GSM8K and MATH problems, where the solutions contain natural language, code and execution results. This dataset is used to train MathCoder-Initial. Afterward, the authors design problem interpolation prompting to generate new training problems with the difficulty between GSM8K and MATH, and use MathCoder-Initial to generate solutions for further finetuning. They evaluate MathCoder on several math benchmarks, and show that MathCoder outperforms other open-source LLMs.

Reviewers raised problems with novelty and with distilling GPT-4 (possible contamination and dependency on proprietary model). This general approach has been shown to be effective at improving the targeted benchmark in many domains and also the same domain (GSM8K and MATH in simultaneous work). However, this approach is also known to be fairly limited to the domains being targeted.

The majority of reviewers voted weak accept, with no strong dissenting view mentioning other points. The release of the models / dataset is potentially beneficial.

**Justification For Why Not Higher Score:**

reviewers are not enthusiastic. nature of the paper mentioned in metareview.

**Justification For Why Not Lower Score:**

majority of reviewer voted to accept. consistency with other likely accepted papers.

---

### Decision · Program_Chairs · 2024-01-16

Accept (poster)